# End-of-life targeted degradation of DAF-2 insulin/IGF-1 receptor promotes longevity free from growth-related pathologies

**Richard Venz[1], Tina Pekec[2,3], Iskra Katic[3], Rafal Ciosk[3,4,5], Collin Yvès Ewald[1]\***

[1]Eidgenössische Technische Hochschule Zürich, Department of Health Sciences and Technology, Institute of Translational Medicine, Schwerzenbach-Zürich, Switzerland; [2]University of Basel, Faculty of Natural Sciences, Basel, Switzerland; [3]Friedrich Miescher Institute for Biomedical Research, Basel, Switzerland; [4]Institute of Bioorganic Chemistry, Polish Academy of Sciences, Noskowskiego, Poland; [5]University of Oslo, Department of Biosciences, Oslo, Norway

**Abstract** Preferably, lifespan-extending therapies should work when applied late in life without causing undesired pathologies. Reducing insulin/insulin-like growth factor (IGF)-1 signaling (IIS) increases lifespan across species, but the effects of reduced IIS interventions in extreme geriatric ages remains unknown. Using the nematode *Caenorhabditis elegans*, we engineered the conditional depletion of the DAF-2/insulin/IGF-1 transmembrane receptor using an auxin-inducible degradation (AID) system. This allowed for the temporal and spatial reduction in DAF-2 protein levels at time points after which interventions such as RNAi become ineffective. Using this system, we found that AID-mediated depletion of DAF-2 protein surpasses the longevity of *daf-2* mutants. Depletion of DAF-2 during early adulthood resulted in multiple adverse phenotypes, including growth retardation, germline shrinkage, egg retention, and reduced brood size. By contrast, AID-mediated depletion of DAF-2 post-reproduction, or specifically in the intestine in early adulthood, resulted in an extension of lifespan without these deleterious effects. Strikingly, at geriatric ages, when 75 % of the population had died, AID-mediated depletion of DAF-2 protein resulted in a doubling in lifespan. Thus, we provide a proof-of-concept that even close to the end of an individual's lifespan, it is possible to slow aging and promote longevity.

**\*For correspondence:**
collin-ewald@ethz.ch

**Competing interest:** The authors declare that no competing interests exist.

## Introduction

The goal of aging research or geroscience is to identify interventions that promote health during old age (*Kennedy et al., 2014*; *López-Otín et al., 2013*; *Partridge et al., 2018*). Nutrient-sensing pathways that regulate growth and stress resistance play major roles as conserved assurance pathways for healthy aging (*Kenyon, 2010*; *López-Otín et al., 2013*). One of the first longevity pathways discovered was the insulin/insulin-like growth factor (IGF)-1 signaling pathway (reviewed in *Kenyon, 2010*). Reducing insulin/IGF-1 signaling (IIS) increases lifespan across species (*Kenyon, 2010*). Mice heterozygous for the IGF-1 receptor, or with depleted insulin receptor in adipose tissue, are stress-resistant and long-lived (*Blüher et al., 2003*; *Holzenberger et al., 2003*), for example, and several single-nucleotide polymorphisms in the IIS pathway have been associated with human longevity (*Kenyon, 2010*). Moreover, gene variants in the IGF-1 receptor have been associated and functionally linked with long lifespans in human centenarians (*Suh et al., 2008*). This suggests that a comprehensive understanding of this pathway in experimental, genetically tractable organisms has promising

**eLife digest** The goal of geroscience, or research into old age, is to promote health during old age, and thus, to increase lifespan. In the body, the groups of biochemical reactions, or 'pathways', that allow an organism to sense nutrients, and regulate growth and stress, play major roles in ensuring healthy aging. Indeed, organisms that do not produce a working version of the insulin/IGF-1 receptor, a protein involved in one such pathway, show increased lifespan. In the worm *Caenorhabditis elegans*, mutations in the insulin/IGF-1 receptor can even double their lifespan. However, it is unclear whether this increase can be achieved once the organism has reached old age.

To answer this question, Venz et al. genetically engineered the nematode worm *C. elegans* so that they could trigger the rapid degradation of the insulin/IGF-1 receptor either in the entire organism or in a specific tissue. Venz et al. started by aging several *C. elegans* worms for three weeks, until about 75% had died. At this point, they triggered the degradation of the insulin/IGF-1 receptor in some of the remaining worms, keeping the rest untreated as a control for the experiment.

The results showed that the untreated worms died within a few days, while worms in which the insulin/IGF-1 receptor had been degraded lived for almost one more month. This demonstrates that it is possible to double the lifespan of an organism at the very end of life.

Venz et al.'s findings suggest that it is possible to make interventions to extend an organism's lifespan near the end of life that are as effective as if they were performed when the organism was younger. This sparks new questions regarding the quality of this lifespan extension: do the worms become younger with the intervention, or is aging simply slowed down?

translational value for promoting health in elderly humans. However, whether or not reducing IIS during end-of-life stages can still promote health and longevity in any organism is unknown. Therefore, we turned to the model organism *Caenorhabditis elegans* to investigate whether reducing IIS during old age was sufficient to increase lifespan.

The groundbreaking discovery that a single mutation in *daf-2*, which is the orthologue of both the insulin and IGF-1 receptors (*Kimura et al., 1997*), or mutations in 'downstream' genes in the IIS pathway, could double the lifespan of an organism was made in the nematode *C. elegans* (*Friedman and Johnson, 1988*; *Kenyon et al., 1993*). Since its discovery, over 1000 papers on *daf-2* have been published, making it one of the most studied genes in this model organism (Source: PubMed). Genetic and genomics approaches have revealed that the DAF-2 insulin/IGF-1 receptor signaling regulates growth, development, metabolism, inter-tissue signaling, immunity, stress defense, and protein homeostasis, including extracellular matrix remodeling (*Ewald et al., 2015*; *Gems et al., 1998*; *Kimura et al., 1997*; *Murphy and Hu, 2013*; *Wolkow et al., 2000*). Much of our knowledge of the effects of *daf-2* on aging has come from the study of reduction-of-function alleles of *daf-2*. Several alleles of *daf-2* have been isolated that are temperature-sensitive with respect to an alternative developmental trajectory. For instance, most *daf-2* mutants develop into adults at 15°C and 20°C but enter the dauer stage at 25 °C (*Gems et al., 1998*), which is a facultative and alternative larval endurance stage in which *C. elegans* spends most of its life cycle in the wild (*Hu, 2007*). Under favorable conditions, *C. elegans* develops through four larval stages (L1–L4). By contrast, when the animals are deprived of food and experience an overcrowded environment and/or thermal stress (above 27 °C), the developing larvae molt into an alternative pre-dauer (L2d) stage. If conditions do not improve, *C. elegans* enter the dauer diapause instead of the L3 stage (*Golden and Riddle, 1984*; *Hu, 2007*; *Karp, 2018*).

A major limitation in using *daf-2* mutants is that several of them show L1 larval and pre-dauer stage (L2d) arrest (*Gems et al., 1998*). Furthermore, the *daf-2* alleles have been categorized into two mutant classes depending on the penetrance of dauer-like phenotypes during adulthood, such as reduced brood size, small body size, and germline shrinkage, as observed in the *daf-2* class II mutants (*Arantes-Oliveira et al., 2003*; *Ewald et al., 2018*; *Ewald et al., 2015*; *Gems et al., 1998*; *Hess et al., 2019*; *Patel et al., 2008*; *Podshivalova and Kerr, 2017*). RNA interference of *daf-2* can be applied, which increases lifespan without dauer formation during development and circumvents induction of *daf-2* class II mutant phenotypes during adulthood (*Dillin et al., 2002*; *Ewald et al., 2018*; *Ewald et al., 2015*; *Kennedy et al., 2004*). However, the increase in lifespan by RNAi of *daf-2* is only partial compared to strong alleles such as *daf-2(e1370)* (*Ewald et al., 2015*). Furthermore,

adult-specific RNAi knockdown of *daf-2* quickly loses its potential to increase lifespan and does not extend lifespan when started after day 6 of adulthood (*Dillin et al., 2002*), that is, after the reproductive period of *C. elegans*. Whether this is due to age-related functional decline of RNAi machinery or residual DAF-2 protein levels, or whether the late-life depletion of *daf-2* simply does not extend lifespan remains unclear. As such, using an alternative method to reduce DAF-2 levels beyond RNAi or *daf-2* mutation may allow us to more clearly uncouple the pleiotropic effects of reduced IIS during development from those that drive *daf-2*-mediated longevity during late adulthood.

To this end, we used an auxin-inducible degradation (AID) system to induce the depletion of the degron-tagged DAF-2 protein with temporal precision (*Zhang et al., 2015*). The *Arabidopsis thaliana* IAA17 degron is a 68-amino acid motif that is specifically recognized by the transport inhibitor response 1 (TIR1) protein only in the presence of the plant hormone auxin (indole-3-acetic acid; *Dharmasiri et al., 2005*). Although cytoplasmic, nuclear, and membrane-binding proteins tagged with degron have been recently shown to be targeted and degraded in *C. elegans* (*Beer et al., 2019*; *Zhang et al., 2015*), to our knowledge, the AID system has not been used previously to degrade transmembrane proteins, such as the DAF-2 insulin/IGF-1 receptor. We find that using AID effectively degrades DAF-2 protein and promotes dauer formation when applied early in development. Dauer-like phenotypes are present in adults when AID of DAF-2 is applied late in development. Some of these adulthood dauer traits are induced by the loss of *daf-2* in neurons, but others seem to be caused by the systemic loss of *daf-2*. More importantly, the post-developmental, conditional degradation of DAF-2 protein extends lifespan without introducing dauer-like phenotypes. Remarkably, we demonstrate that when more than half of the population has died at day 25 of adulthood, AID of DAF-2 in these remaining aged animals is sufficient to promote longevity. Our work suggests that therapeutics applied at even extremely late stages of life are capable of increasing longevity and healthspan in animals.

## Results

### Generation and validation of a degron-tagged DAF-2 receptor

To monitor and conditionally regulate protein levels of the *C. elegans* DAF-2 insulin/IGF-1 receptor, we introduced a degron::3xFLAG tag into the 3' end of the *daf-2* open reading frame (*Figure 1—figure supplement 1*). This degron::3xFLAG insertion into the genome was designed to tag the DAF-2 receptor at the cytosolic part for two reasons: first, to minimize any interference by the 81-amino acids large degron::3xFLAG-tag with the DAF-2 receptor function; and second, to ensure accessibility of the degron for targeted degradation by the TIR1 ubiquitin ligase expressed in the cytoplasm (*Figure 1A*). We endogenously tagged the DAF-2 receptor using CRISPR, and the resulting *daf-2(bch40)* CRISPR allele was verified by PCR (*Figure 1—figure supplement 1*). We performed western blot analysis against the 3xFLAG-tag and detected a specific band in *daf-2(bch40)* animals. This band was absent in wild type (N2) and animals carrying only the *eft-3p*::TIR1::mRuby::*unc-54* 3'UTR transgene (*Figure 1B*), which expresses TIR1 in all somatic cells, including neurons (*Tomioka et al., 2016*). To promote degradation of the degron::3xFLAG-tagged DAF-2 receptor, we crossed *daf-2(bch40)* into TIR1-expressing *C. elegans*. The strain obtained from this cross will be called 'DAF-2::degron' throughout this paper (i.e., *Si57* [P*eft-3*::TIR1::mRuby::*unc-54* 3'UTR+ *Cbr-unc-119*(+)] II; *daf-2(bch40)* [degron::3xFLAG::STOP::SL2-SV40-degron::wrmScarlet-*egl-13* NLS]) III). This strain showed no obvious phenotypes and exhibited a normal developmental progression at 20 °C (*Figure 1—figure supplement 1*). To verify whether the band from the western blot was indeed DAF-2::degron::3xFLAG, we treated DAF-2::degron animals with *daf-2* RNAi. The band nearly completely disappeared after 48 hr of *daf-2(RNAi)* feeding (*Figure 1C and D*, *Source data 1* and *Source data 2*). Collectively, these results suggested that the tagged transmembrane receptor DAF-2 did not interfere with normal DAF-2 function.

### Dietary changes modulate endogenous DAF-2/insulin/IGF-1 receptor abundance

Next, we monitored endogenous DAF-2 protein levels under different environmental conditions, such as temperature and diet. Previously, Kimura and colleagues used DAF-2 antibody immunostaining of whole animals and reported that mutant DAF-2(*e1370*) protein is present at 15 °C but barely detected at 25 °C, whereas mutant DAF-2(*e1370*) protein in a *daf-16* null background or wild-type

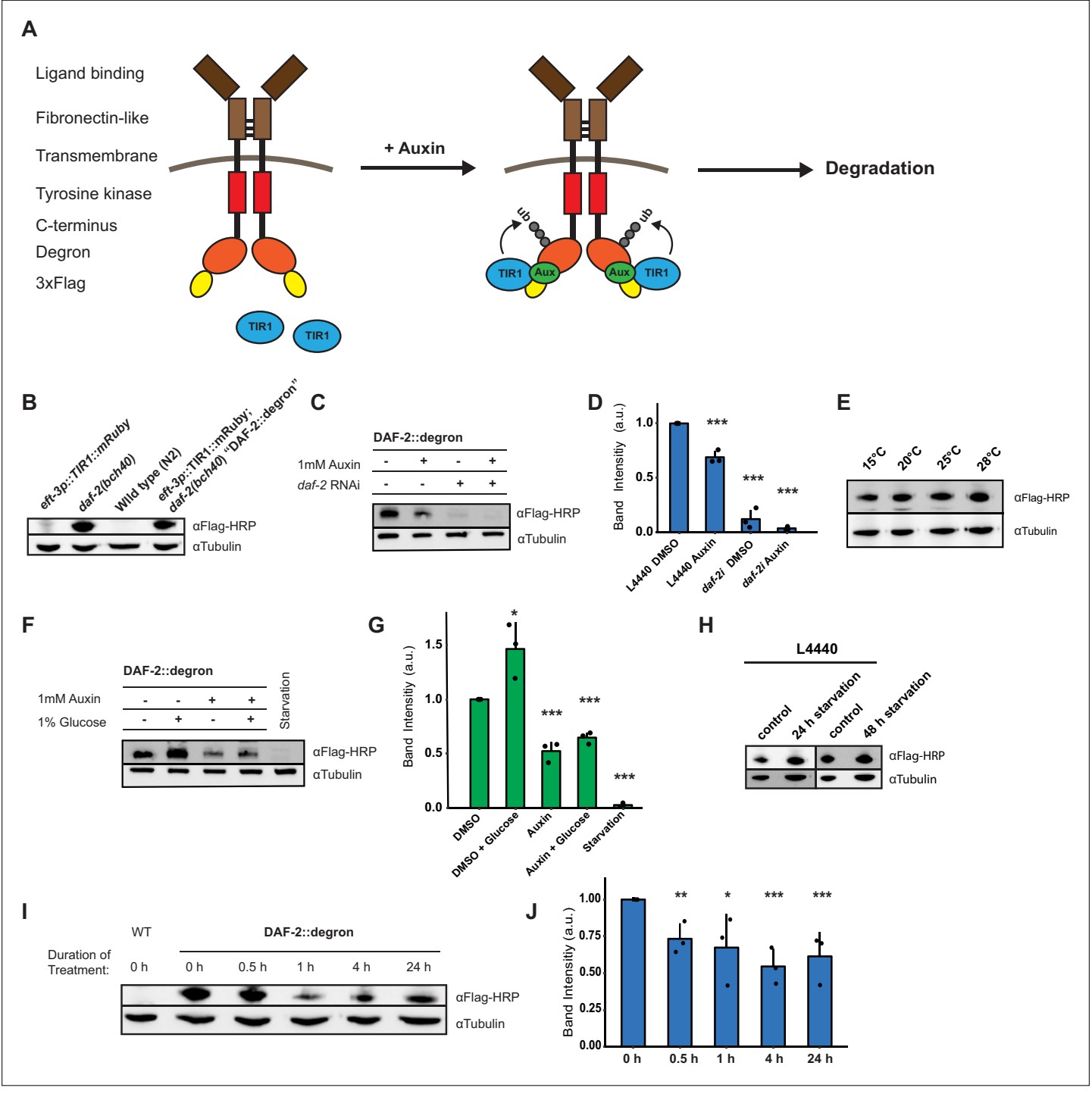

**Figure 1.** Degron-tagged transmembrane insulin/insulin-like growth factor-1 (IGF-1) receptor DAF-2 is susceptible to auxin-mediated degradation. (**A**) Schematic illustration of auxin-inducible degradation (AID)-mediated DAF-2 receptor depletion in *daf-2(bch40) Caenorhabditis elegans*. (**B**) Immunoblot of *eft-3p::TIR1::mRuby::unc-54* 3'UTR, *daf-2(bch40)*, wild type (**N2**), and DAF-2::degron (*eft-3p::TIR1::mRuby::unc-54* 3'UTR; *daf-2(bch40)*). (**C**) Immunoblot of 'DAF-2::degron' animals that were grown on OP50 NGM and at L4 stage shifted to either empty vector control RNAi (L4440) or *daf-2*(RNAi) plates containing either DMSO or 1 mM auxin. After 48 hr on the second day of adulthood, animals were harvested for western blotting. (**D**) Densitometric quantification of (**C**) from n = 3 independent experiments. Error bars represent s.d. Two-sided *t*-test was used for statistical analysis. *: p < 0.05, **: p < 0.01, ***: p < 0.001. (**E**) Immunoblot of DAF-2::degron animals showed no decrease of DAF-2 levels at high temperatures. Animals were raised at 15 °C and put as L4 for 24 hr at the indicated temperatures. (**F**) A representative immunoblot of 'DAF-2::degron' animals after 1 % glucose and 36–48 hr starvation on the second day of adulthood. L4 DAF-2::degron animals were either placed on OP50 NGM plates with or without 1 mM auxin, or containing 1 % glucose, or on empty (no bacteria) NGM plates. Animals were harvested 36–48 hr later. (**G**) Densitometric quantification

*Figure 1 continued on next page*

*Figure 1 continued*

of (**F**) from n = 3 independent experiments. Error bars represent s.d. Two-sided *t*-test was used for statistical analysis. *: p < 0.05, **: p < 0.01, ***: p < 0.001. (**H**) Immunoblot analysis of starved DAF-2::degron animals. Animals were raised on OP50 NGM at 20 °C and shifted from L4 to L4440 containing FUdR. After 2 days, they were washed off, and either frozen as control or put on empty plates and harvested after 24 or 48 hr, respectively. (**I**) Immunoblot analysis of 1-day-old adult DAF-2::degron animals treated with 1 mM auxin for the indicated time periods. (**J**) Quantification of (**I**) from n = 3 independent experiments. Error bars represent s.d. One-sided *t*-test was used for statistical analysis. *: p < 0.05, **: p < 0.01, ***: p < 0.001. For (**B–J**), see *Source data 1* and *Source data 2* for raw data, full blots, and statistics.

The online version of this article includes the following figure supplement(s) for figure 1:

**Figure supplement 1.** Degron-tagged DAF-2 is functional and susceptible to auxin-mediated degradation.

DAF-2 protein persists at both 15°C and 25°C (*Kimura et al., 2011*). By contrast, upon 24 hr of starvation, the DAF-2 receptor is no longer detectable by using immunofluorescence in fixed *C. elegans* (*Kimura et al., 2011*). Since the FOXO transcription factor *daf-16* is the transcriptional output of *daf-2* signaling (*Ewald et al., 2018*; *Gems et al., 1998*), these results suggest that DAF-2 protein levels may be autoregulated by IIS and might be influenced by temperature and food availability. We first asked whether our DAF-2::degron::3xFLAG tag allows quantification of endogenous DAF-2 levels. We observed comparable wild-type DAF-2::degron::3xFLAG levels across a range of temperatures (15–28°C; *Figure 1E*), indicating that temperature does not influence DAF-2 levels in wild type. Intriguingly, however, we found that using FLAG-HRP antibodies to monitor protein levels, DAF-2 protein almost completely disappeared after 36–48 hr of starvation (*Figure 1F and G*). In keeping with this result, well-fed animals, for which we added 1 % glucose into the bacterial diet (OP50), increased the DAF-2 protein levels (*Figure 1F and G*). Curiously, we noted that this starvation-induced degradation of DAF-2 did not happen when, during development, *C. elegans* were fed another bacterial strain, HT1115, used for RNAi (L4440). When DAF-2::degron animals were grown on L4440 and then shifted on empty NGM plates for 24 or 48 hr of starvation, DAF-2 levels did not decrease (*Figure 1H*). This observation suggests a hypothesis that the nutritional composition of the animal's diet prior to starvation influences DAF-2 stability, which will be interesting to test in future research. We conclude that food availability controls not only the secretion of insulin-like peptides to regulate DAF-2 activity (*Pierce et al., 2001*) but also DAF-2 receptor abundance.

## Auxin-induced degradation of degron-tagged insulin/IGF-1 receptor

In *C. elegans*, cytosolic degron-tagged proteins are almost completely degraded after 30 min of auxin treatment (*Zhang et al., 2015*). However, the degradation of transmembrane proteins using AID in vivo has not been previously reported. We hypothesized that *C. elegans* might exhibit similar kinetics of degradation of a transmembrane protein following auxin treatment. In keeping with that hypothesis, after 30 min of 1 mM auxin treatment, we observed a dramatic decrease in transmembrane DAF-2 protein abundance (*Figure 1I and J*). Levels of DAF-2 were only slightly further reduced by continued auxin treatment, as indicated at 4 and 24 hr time points (*Figure 1I and J*). After 24 hr of 1 mM auxin treatment, we observed only a 40 % total decrease in DAF-2 protein abundance rather than a complete loss (*Figure 1C and D*). Similar kinetics in the degradation of DAF-2::degron::3xFLAG levels were confirmed using additional FLAG and degron antibodies (*Source data 2*). Taken together, these results suggest that our AID system allows for the partial, rapid degradation of the transmembrane DAF-2 receptor.

## Inactivation of DAF-2::degron by the AID inhibits downstream IIS

We wondered whether the reduction of DAF-2 levels by AID would have consequences consistent with reduced IIS. Activation of DAF-2/insulin/IGF-1 receptor induces a downstream kinase cascade to phosphorylate the transcription factors DAF-16/FOXO and SKN-1/NRF, causing their retention in the cytoplasm (*Figure 2A*; *Ewald et al., 2015*; *Henderson and Johnson, 2001*; *Lin et al., 2001*; *Murphy et al., 2003*; *Ogg et al., 1997*; *Tullet et al., 2008*). Genetic inhibition of *daf-2* results in less DAF-16 and SKN-1 phosphorylation and promotes nuclear translocation to induce the expression of target genes, such as *sod-3* (superoxide dismutase) and *gst-4* (glutathione S-transferase), respectively (*Ewald et al., 2015*; *Henderson and Johnson, 2001*; *Lin et al., 2001*; *Murphy et al., 2003*; *Tullet et al., 2008*). Within 1 hr of 1 mM auxin treatment, we found that most DAF-16::GFP translocated into the nuclei in DAF-2::degron animals (*Figure 2B*), with observable translocation already after

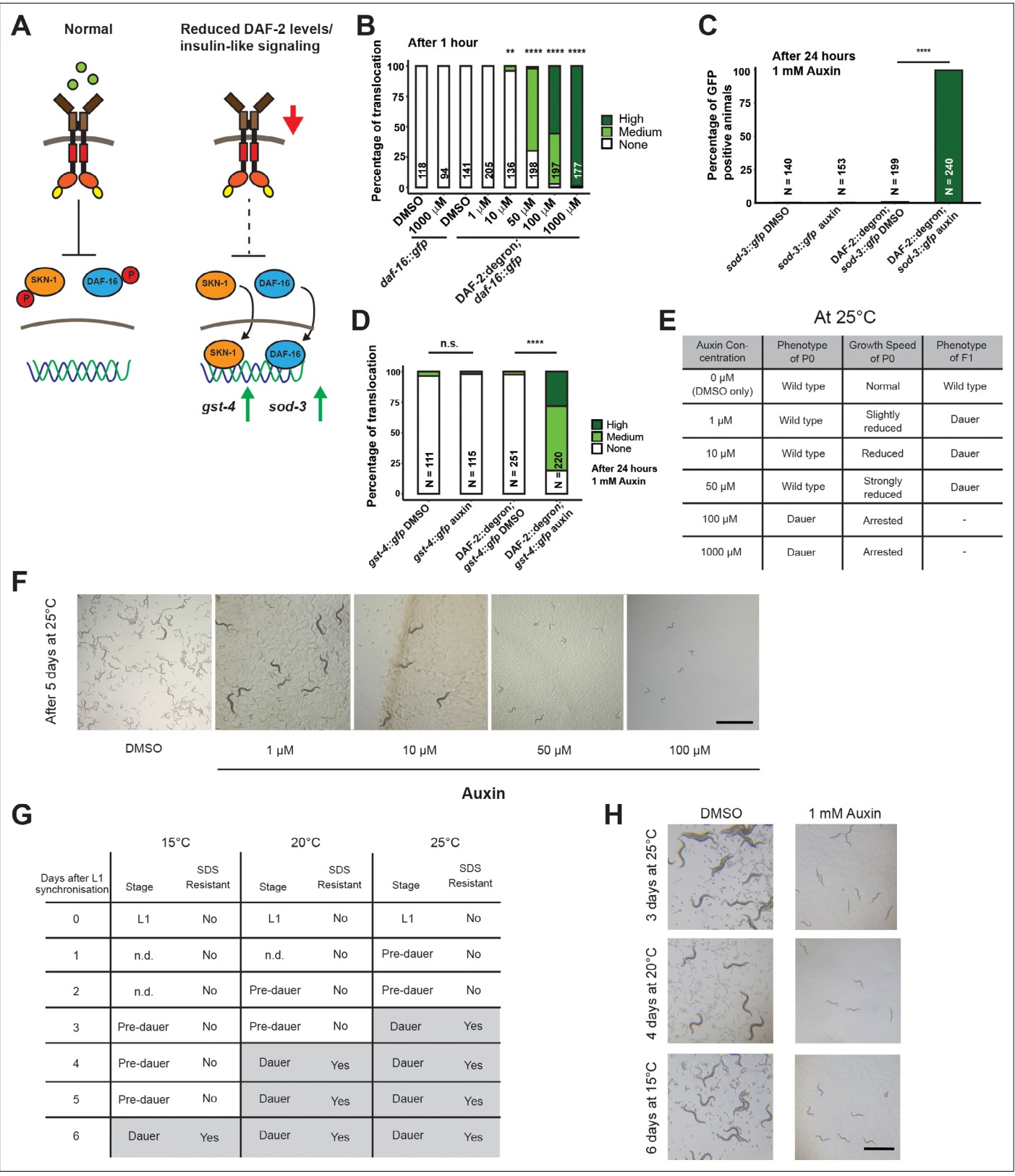

**Figure 2.** Inactivation of DAF-2::degron by the auxin-inducible degradation (AID) upregulated downstream reporters and caused dauer entry at any temperature. (**A**) A schematic illustration of the DAF-2 signaling pathway. DAF-2 phosphorylates the transcription factors DAF-16 and SKN-1 through a cascade of kinases and sequesters them to the cytosol. Lower DAF-2 levels or activity leads to de-phosphorylation and translocation of DAF-16 and SKN-1 to the nucleus and expression of target genes like *sod-3* and *gst-4*, respectively. (**B**) One hour exposure to auxin led to nuclear translocation of

*Figure 2 continued*

DAF-16::GFP in a concentration-dependent manner in DAF-2::degron; *daf-16::gfp* animals, but not in *daf-16::gfp* at the L4 stage. n > 94 from two (for *daf-16::gfp*) or three (for DAF-2::degron; *daf-16::gfp*) independent experiments. **: p < 0.01, ****: p < 0.0001. (**C**) Exposure to 1 mM auxin activates the reporter *sod-3::GFP* in DAF-2::degron; *sod-3::gfp* but not in animals that carry *sod-3::gfp* alone. L4 animals were exposed to 1 mM auxin or DMSO for 24 hr and GFP levels were scored (i.e., at day 1 of adulthood). n > 140 from two (for *sod-3::gfp*) or four (for DAF-2::degron; *sod-3::gfp*) independent experiments. ****: p < 0.0001. (**D**) Exposure to 1 mM auxin activates the reporter *gst-4::GFP* in DAF-2::degron; *gst-4::gfp* but not in animals that carry *gst-4::gfp* alone. L4 animals were exposed to 1 mM auxin or DMSO for 24 hr and GFP levels were scored (i.e., at day 1 of adulthood). n > 111 from two (for *gst-4::gfp*) or four (for DAF-2::degron; *gst-4::gfp*) independent experiments. ****: p < 0.0001. (**E**) Auxin treatment of DAF-2::degron affected development. Synchronized L1 put at low concentrations (1–50 μM auxin) shows reduced growth speed, and their offspring enters dauer. At high concentrations (100 and 1000 μM), the L1 animals arrest and enter the dauer stage after a few days. (**F**) Representative pictures of growth impairment caused by auxin-mediated degradation of DAF-2 in DAF-2::degron animals at different concentrations. Bar = 1 mm. (**G**) Dauer entry of DAF-2::degron animals at 1 mM auxin was temperature-independent, but the time needed for dauer entry is temporally scaled. To distinguish dauer animals from pre-dauer animals, they were treated for 15 min with 1 % SDS. Only dauer animals survived SDS treatment. (**H**) Microscope pictures of DAF-2::degron animals after dauer entry (right column) and control animals (left column) kept for the same time at 15 °C, 20 °C, and 25 °C. Control animals were on their second day of adulthood when the auxin-treated counterparts entered the dauer stage. Bar = 1 mm. For (B–D), see *Source data 1* for raw data and statistics.

The online version of this article includes the following figure supplement(s) for figure 2:

**Figure supplement 1.** Inactivation of DAF-2::degron by the auxin-inducible degradation (AID) upregulated downstream reporters and caused dauer entry at any temperature.

**Figure supplement 2.** Depletion of DAF-2::degron at mid-L1 promoted *daf-16*-dependent dauer entry.

30 min (*Figure 2—figure supplement 1*). This DAF-16::GFP nuclear localization in DAF-2::degron animals was time- and auxin-concentration-dependent and did not occur in DAF-16::GFP animals with wild-type DAF-2 (*Figure 2B*, *Figure 2—figure supplement 1*). Similarly, SKN-1- or DAF-16-target gene expression of *gst-4* or *sod-3* was only induced upon auxin treatment in DAF-2::degron animals (*Figure 2C and D*). Thus, expectedly, IIS is reduced upon AID DAF-2 degradation.

## AID of DAF-2::degron promotes dauer entry at any temperature

Reduced IIS during development promotes dauer entry. Dauer formation at 15 °C has been observed for a variety of strong loss-of-function *daf-2* alleles, such as the class I alleles *e1369* and *m212*, the class II allele *e979*, the null alleles *m65*, *m646*, *m633*, and a variety of unclassified alleles discovered by Malone and Thomas (*Gems et al., 1998*; *Kimura et al., 2011*; *Malone and Thomas, 1994*; *Patel et al., 2008*). For the commonly used reference alleles *e1368* and *e1370*, penetrant dauer formation only occurs at 25 °C (*Gems et al., 1998*). By contrast, knockdown of *daf-2* by RNAi does not cause dauer formation at any temperature (*Dillin et al., 2002*; *Ewald et al., 2015*; *Kennedy et al., 2004*). We hypothesized that dauer formation would not happen because the decrease of DAF-2::degron levels after auxin treatment is only around 40 %. However, synchronized L1 treated with 0.1 or 1 mM auxin all formed dauers at 25 °C (*Figure 2E and F*). We observed dose-dependent retardation of the developmental speed when using lower auxin concentrations (1, 10, and 50 μM), but the offspring of retarded animals, grown on plates containing auxin, became dauers (*Figure 2E and F*). The dependence on auxin concentration for dauer formation of L1 animals suggests a threshold of DAF-2 receptor levels for the decision or commitment to dauer diapause. Even more surprising was the observation that dauer formation was also observed at 15°C and 20°C with complete penetrance (*Figure 2G and H*). We found that dauer formation was related to the developmental speed at a given temperature: At 15 °C, it took 6 days; at 20 °C, it took 4 days; and at 25 °C, it took 3 days to form dauers (*Figure 2H*). We verified that all auxin-induced DAF-2::degron dauers showed dauer-specific characteristics, such as SDS resistance, cessation of feeding, constricted pharynxes, and dauer-specific alae (*Figure 2G*, *Figure 2—figure supplement 1*), suggesting a complete dauer transformation. Thus, the AID of DAF-2 promotes complete dauer formation independent of temperature but dependent on DAF-2 protein abundance.

## Dauer commitment at mid-larval stage 1 upon DAF-2 degradation

Wild-type animals enter the pre-dauer L2d stage, where they keep monitoring their environment before completely committing to dauer formation (*Golden and Riddle, 1984*; *Hu, 2007*; *Karp, 2018*). Treating wild type with dauer pheromone suggested mid-L1 as the stage when the dauer decision is

made (*Golden and Riddle, 1984*). By contrast, previous temperature-shifting experiments (from 15°C to 25°C) with *daf-2* mutants suggested a dauer decision time window from L1 to L2 stage, before the L2d stage (*Swanson and Riddle, 1981*). Since AID allows for precise temporal degradation of DAF-2, we pinpointed the dauer entry decision to the mid-L1 stage. Specifically, we shifted synchronized L1s at different time points to plates containing 1 mM auxin and counted the number of cells in developing gonads to determine the developmental stage, when 50 % of the population committed to becoming dauers (*Figure 3—figure supplement 1*). We found that when DAF-2 levels are below a given threshold at the mid-L1 stage, the animals commit to becoming dauers.

## AID-degraded DAF-2 resembles a non-conditional and severe loss-of-function *Daf-2* allele

The FOXO transcription factor DAF-16 is required for dauer formation in *daf-2* mutants. We crossed DAF-2::degron with DAF-16::degron (*Aghayeva et al., 2021*) and found that *daf-16* was required for dauer formation and developmental speed alterations after DAF-2::degron depletion (*Figure 2—figure supplement 2*). Previous reports suggest that many *daf-2* alleles show low to severe penetrance of embryonic lethality and L1 arrest at higher temperatures (*Collins et al., 2008*; *Ewald et al., 2016*; *Gems et al., 1998*; *Patel et al., 2008*). Although the constitutive dauer formation of the proposed null allele *daf-2(m65)* is suppressed by *daf-16* null mutations, the embryonic lethality and L1 arrest are not *daf-16*-dependent (*Patel et al., 2008*). We observed no embryonic lethality nor L1 arrest in the progeny of animals placed on 1 mM auxin as L4s. Similar results were seen using either the DAF-2::degron or DAF-2::degron; germline TIR1 strains. Higher concentrations of auxin lead to toxicity in both wild-type and DAF-2::degron animals (*Figure 2—figure supplement 2*). A lack of embryonic lethality could be explained by insufficient DAF-2 degradation or earlier decision stages. Taken together, the inactivation of DAF-2 by AID is 100 % penetrant for dauer formation at any temperature. Still, the absence of embryonic lethality or L1 arrest at 1 mM auxin suggests that DAF-2::degron functionally is more similar to a non-conditional and severe loss-of-function mutation than a null allele.

## Enhanced lifespan extension by AID of DAF-2 in adult animals

Given the strong phenotypic effects of DAF-2 AID on animal development, we next explored whether DAF-2 degradation by AID could affect the function of adult animals. Previous studies indicate that reducing IIS, either by *daf-2* RNAi knockdown or in genetic mutants, increases lifespan at any temperature (15–25°C) (*Ewald et al., 2018*; *Gems et al., 1998*). We hypothesized that AID-dependent degradation of DAF-2 would have similar effects on the lifespan of animals. We found that auxin supplementation of DAF-2::degron animals, starting from L4, resulted in a 70–135% lifespan extension (*Figure 3A*; *Supplementary file 1*). Impressively, DAF-2 degradation using 1 mM auxin surpassed the longevity of commonly used *daf-2(e1368)* and *daf-2(e1370)* mutants (*Figure 3A*, *Supplementary file 1*). By contrast, auxin treatment at 0.1 or 1 mM concentration had little or no effect on wild-type lifespan (*Figure 3A*; *Supplementary file 1*). These results suggest that auxin-induced degradation of *daf-2* is a powerful tool to promote longevity.

## Manifestation of some *Daf-2* class II mutant phenotypes during adulthood at 15°C without passing through L2d

Although reducing *daf-2* function causes beneficial increases in longevity and stress resistance, it causes residual detrimental phenotypes in adult animals that resemble the behavioral and morphologic changes reminiscent of developing animals remodeling to enter the dauer state (*Ewald et al., 2018*; *Gems et al., 1998*). In class II *daf-2* alleles, these phenotypes, such as small gonads, reduced brood size, reduced motility, and reduced brood size, manifest only at 25 °C during adulthood but not at lower temperatures (*Ewald et al., 2018*; *Gems et al., 1998*). To determine whether DAF-2::degron AID animals display *daf-2* class II mutant phenotypes, we quantified these characteristics at 15°C and 25°C. In placing L4 DAF-2::degron animals on auxin and at 25 °C, we observed similar levels of egg retention and effects on gonad size as was seen in the *daf-2(e1370)* class II allele. Strikingly, these effects were temperature-dependent, as DAF-2::degron animals did not retain eggs or had reduced gonad sizes at 15 °C (*Figure 3B*, *Figure 3—figure supplement 1*). Similarly, DAF-2::degron animals on auxin exhibited germline shrinkage at 25 °C, albeit to a lesser degree than the *daf-2(e1370)* class

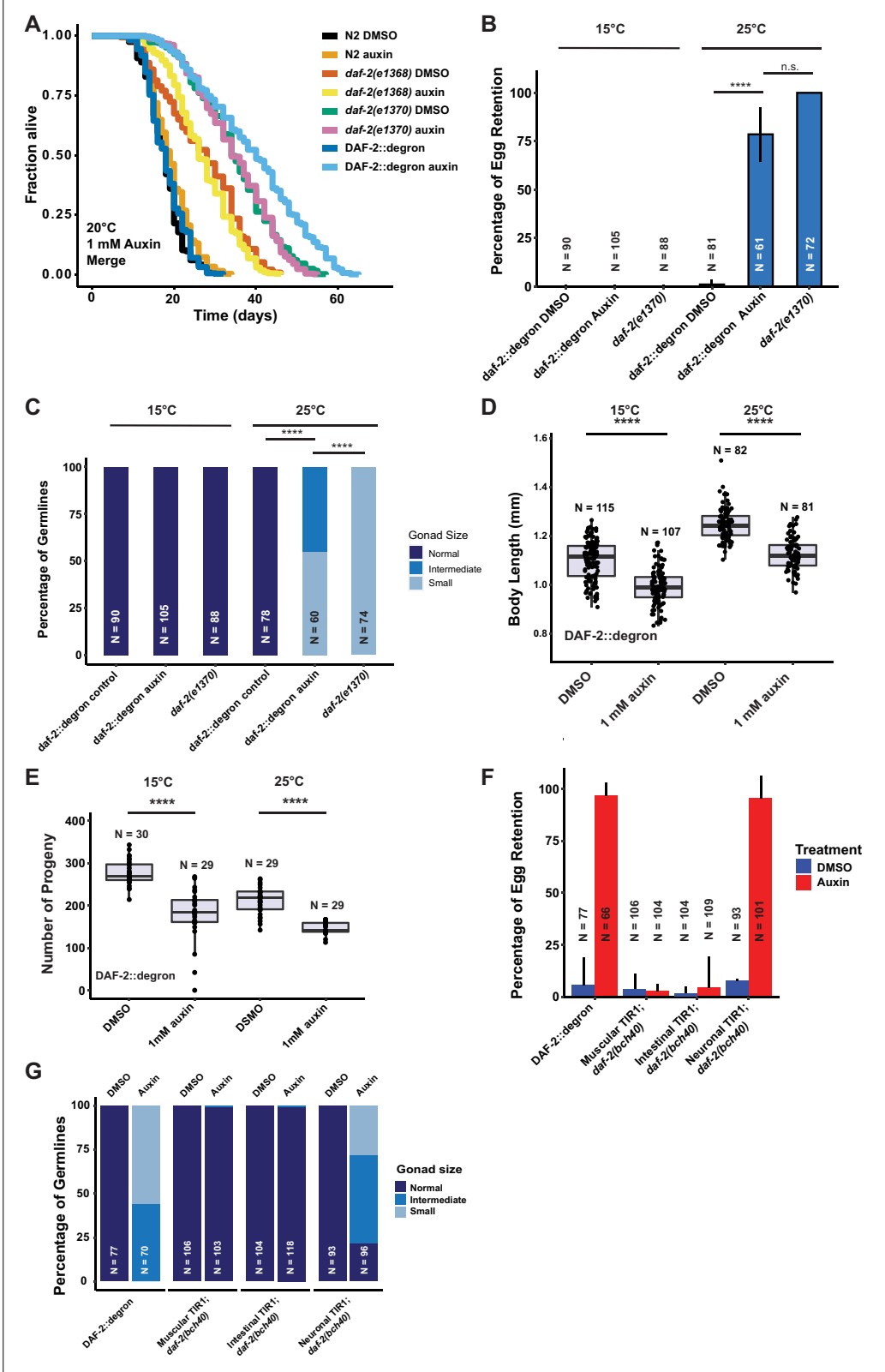

**Figure 3.** Depletion of DAF-2::degron resulted in *daf-2* class II mutant phenotypes at 15 °C. (**A**) Auxin treatment extended the lifespan of DAF-2::degron animals. Animals were shifted as L4 to plates containing DMSO or 1 mM auxin at 20 °C. (**B**) 1 mM auxin treatment leads to the 'dauer-like' egg retention phenotype in DAF-2::degron animals. The animals were raised at 15 °C, then shifted as L4 to 25 °C or kept at 15 °C. Two days later at 25 °C and

*Figure 3 continued on next page*

*Figure 3 continued*

3 days later at 15 °C, they were examined for egg retention. The experiment was performed three independent times. Error bar represents s.d. ****: p < 0.0001. (**C**) Gonads were shrunk after 1 mM auxin treatment in DAF-2::degron animals. Animals were raised at 15 °C and shifted as L4 to 25 °C or kept at 15 °C. Two days later at 25 °C and 3 days later at 15 °C, the animals were checked for gonad size. The experiment was performed three independent times. ****: p < 0.0001. (**D**) 1 mM auxin treatment decreased the body size of DAF-2::degron animals at 15°C and 25°C. Animals were raised at 15 °C and shifted to 1 mM auxin or DMSO plates at the L4 stage. The experiment was performed three independent times. ****: p < 0.0001. (**E**) 1 mM auxin treatment of DAF-2::degron resulted in a smaller brood size at 15°C and 25°C. Animals were shifted to 1 mM auxin or DMSO plates at the L4 stage and either kept at 15 °C or moved to 25 °C. The experiment was performed three independent times. ****: p < 0.0001. (**F**) Tissue-specific depletion of DAF-2 in neurons caused egg retention phenotype. Animals were raised at 15 °C and shifted from L4 to 25 °C. Two days later, the animals were checked for egg retention. The experiment was performed three independent times. Error bar represents s.d. ****: p < 0.0001. (**G**) Gonads were shrunk after neuronal depletion of DAF-2. Animals were raised at 15 °C and shifted from L4 to 25 °C. Two days later, the animals were checked for gonad size. The experiment was performed three independent times. ****: p < 0.0001. For (A–G), see *Source data 1* for raw data and *Supplementary file 1* for statistics and additional trials.

The online version of this article includes the following figure supplement(s) for figure 3:

**Figure supplement 1.** Depletion of DAF-2::degron resulted in *daf-2* class II mutant phenotypes at 15 °C.

**Figure supplement 2.** Oxidative resistance and tissue-specific effects after DAF-2 depletion.

II allele (*Figure 3C*, *Figure 3—figure supplement 1*). This phenotype was also absent at 15 °C, suggesting that egg retention and germline shrinkage are temperature-sensitive traits.

Another known *daf-2* class II mutant phenotype at 25 °C is the quiescence or immobility of class II *daf-2(e1370)* mutants (*Ewald et al., 2018*; *Gems et al., 1998*). We did not observe any immobility of auxin-treated DAF-2::degron animals at 25 °C or during lifespan assays at 20 °C (*Video 1*, *Supplementary file 2*). Although the effects on body size of *daf-2(e1370)* class II allele are temperature-dependent, presenting at 25 °C but not at 15 °C (*Ewald et al., 2015*; *Gems et al., 1998*; *McCulloch and Gems, 2003*; *Figure 3—figure supplement 1*), auxin-induced degradation of DAF-2 starting from L4 shortened body size of 2-day-old adults at both temperatures (*Figure 3D*, *Figure 3—figure supplement 1*). Similarly, while *daf-2(e1370)* mutants only exhibit reduced brood sizes at higher temperatures (*Ewald et al., 2018*; *Gems et al., 1998*), AID of DAF-2::degron starting from L4 reduced brood size at both 15°C and 25°C (*Figure 3E*). This suggests that smaller body and brood size manifest as non-conditional traits, in keeping with insulin/IGF-1's role as an essential gene for these functions. In summary, these results suggest that some *daf-2* class II mutant phenotypes, or pathologies, can be induced during adulthood independent of temperature and that passing through the L2d stage is not required for the *daf-2* class II mutant phenotypes to emerge in adult animals.

## Tissue-specific DAF-2 degradation reveals neuronal regulation of egg retention and germline remodeling

The pleiotropic effects of DAF-2 have been ascribed to tissue-specific functions of DAF-2. DAF-2 protein levels are predominantly found in the nervous system and intestine, and to a lesser extent in the hypodermis (*Kimura et al., 2011*), while *daf-2* mRNA expression has also been detected in the germline (*Han et al., 2017*; *Lopez et al., 2013*). Mosaic loss of *daf-2* in different cell lineages indicated neurons as crucial tissue to control dauer formation non-cell autonomously (*Apfeld and Kenyon, 1998*). Moreover, dauer formation in *daf-2(e1370)* can be restored by expressing wild-type DAF-2 only in neurons (*Wolkow et al., 2000*). Thus, we hypothesized that select tissues might drive the *daf-2* class II mutant phenotypes. To test this, we expressed TIR1 specifically in muscles, neurons, and intestine, driven by the *myo-3*, *rab-3*, and *vha-6* promoters, respectively

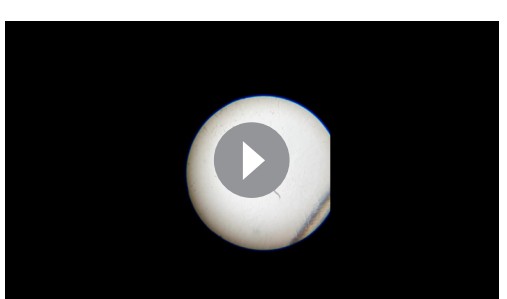

**Video 1.** Video of DAF-2::degron animals treated with 1 mM auxin at day 7 of adulthood at 25 °C. The number corresponds to the different culturing lifespan plates.
https://elifesciences.org/articles/71335/figures#video1

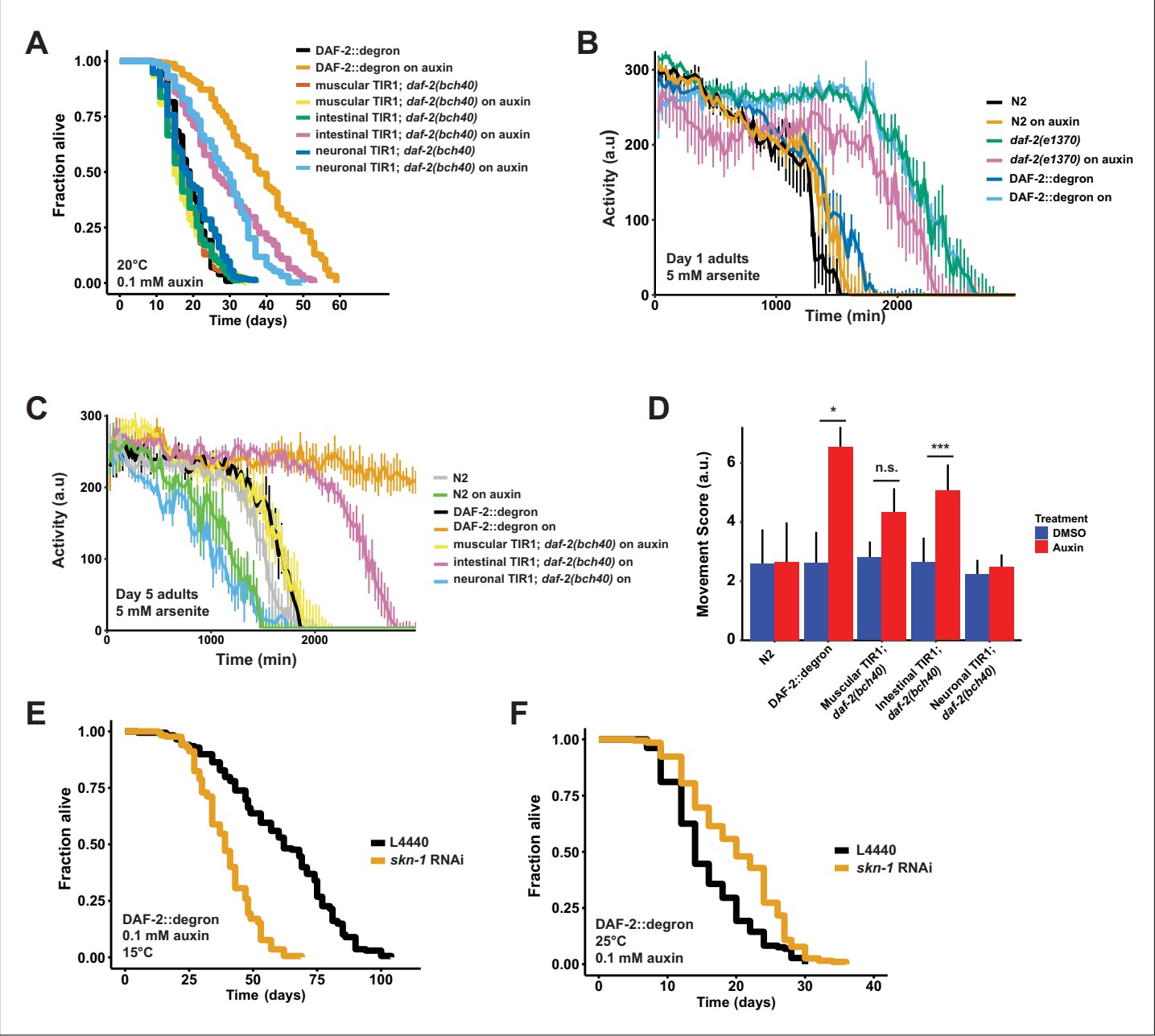

**Figure 4.** Tissue-specific effects after DAF-2 depletion on lifespan extension and reactive oxygen species resistance. (**A**) Auxin treatment extended lifespan in aged DAF-2::degron animals. Animals were shifted from DMSO plates to plates containing 1 mM auxin on the indicated day. (**B**) Auxin-mediated depletion of DAF-2 enhanced oxidative stress resistance to a similar extent as *daf-2(e1370)* animals. Animals were shifted as L4 for 24 hr on plates containing DMSO or 1 mM auxin at 20 °C. (**C**) Intestinal knockdown of DAF-2 enhanced oxidative stress resistance. Animals were shifted to DMSO or 1 mM auxin plates at the L4 stage and kept until day 5 of adulthood at 20 °C. (**D**) Quantification of (**C**) from three independently performed experiments. Error bar represents s.d. *: $p < 0.05$, ***: $p < 0.001$. (**E**) *skn-1* was necessary for full lifespan extension in DAF-2 depleted animals at 15 °C. Animals were kept on 1 mM auxin. $p < 0.0001$. (**F**) Knockdown of *skn-1* extended the mean lifespan of DAF-2 depleted animals at 25 °C. Animals were raised at 15 °C and shifted to 1 mM auxin at 25 °C. $p < 0.0001$. For (**A–F**), see *Source data 1* and *Source data 3* for raw data and *Supplementary file 1* for statistics and additional trials.

(Materials and methods, *Supplementary file 2*). TIR1 expressed from any of these three tissue-specific promoters did not result in reduced body size (*Figure 4E* and *Figure 3—figure supplement 1E*). To validate that the neuronal TIR1 was functional, we crossed neuronal TIR1 into *daf-16(ot853* [*daf-16*::linker::mNG::3xFLAG::AID]) (*Aghayeva et al., 2021*) and observed that DAF-16::mNG was selectively degraded in neurons upon auxin treatment (*Figure 3—figure supplement 2*). Nonetheless, we

found that depletion of DAF-2 in neurons caused egg retention and germline shrinkage (*Figure 3F and G*). Interestingly, the germline shrinkage and egg retention phenotypes were temperature-dependent, suggesting some interaction of temperature and neuronal DAF-2 abundance. Thus, some *daf-2*-phenotypes appear to emerge from a single tissue, whereas others might be due to an interplay between several tissues.

## Tissue-specific AID reveals different requirements for DAF-2 in neurons and intestine for longevity and oxidative stress resistance

Previously, transgenic expression of wild-type copies of DAF-2 in neuronal or intestinal cells was shown to partially suppress the longevity of *daf-2(e1370)* mutants at 25 °C (*Wolkow et al., 2000*). Taking advantage of our unique AID system, we wanted to test whether the degradation of DAF-2 in a single tissue is sufficient to induce longevity. We found that either neuronal or intestinal depletion of DAF-2 was alone sufficient to extend lifespan, although not to the extent as when DAF-2 is degraded in all tissues (*Figure 4A*; *Supplementary file 1*). Therefore, we asked whether tissue-specific DAF-2 degradation was also sufficient for stress resistance seen in *daf-2* mutant animals (*Ewald et al., 2018*; *Gems et al., 1998*). Auxin-induced degradation of DAF-2 in all tissues resulted in increased oxidative stress resistance, comparable to *daf-2(e1370)* mutants (*Figure 4B*). We observed improved oxidative stress resistance when we depleted DAF-2 in the intestine but not in neurons (*Figure 4C and D*, *Figure 3—figure supplement 2*). This implies that reducing DAF-2 levels, specifically in the intestine, promotes longevity and stress resistance without causing dauer-like phenotypes.

## *Skn-1* works in a temperature-sensitive manner but independently of dauer-like reprogramming

We have previously shown that the transcription factor SKN-1/NRF1,2,3 is localized in the nucleus at 15 °C or 25 °C in *daf-2(e1370)* mutants (*Ewald et al., 2015*), suggesting that SKN-1 activation occurs under reduced insulin/IGF-1 receptor signaling conditions (*Tullet et al., 2008*). Intriguingly, *skn-1* activity is necessary for full lifespan extension of *daf-2(e1370)* mutants only at 15 °C but not at 25 °C (*Ewald et al., 2015*). We hypothesized that *skn-1* requirements for lifespan extension are masked when dauer-like reprogramming conditions are triggered at higher temperatures (*Ewald et al., 2018*). Because auxin-treated DAF-2::degron AID animals exhibit *daf-2* class II mutant traits during adulthood at 15 °C, we asked whether the lifespan extension caused by DAF-2::degron AID upon auxin treatment at 15 °C is *skn-1*-independent. We found that the lifespan extension in DAF-2::degron animals fully required *skn-1* at 15 °C. Surprisingly, however, a loss of *skn-1* extended the median lifespan of DAF-2::degron animals at 25 °C (*Figure 4E and F*; *Supplementary file 1*). This suggests that *skn-1* may function independently from dauer-like reprogramming pathways at 15 °C. Furthermore, the increased longevity seen in DAF-2::degron animals may result from a differential transcriptional program at higher temperatures compared to lower temperatures.

## Late-life application of AID of DAF-2 increases lifespan

Finally, we asked whether it would be possible to promote longevity in geriatric animals by depleting DAF-2 by AID. Previous studies using RNAi indicated that reduced *daf-2* expression extended lifespan when started at day 6 of adulthood but not later (*Dillin et al., 2002*), raising the question of whether *daf-2*-longevity induction is possible beyond the reproductive period (days 1–8 of adulthood). To address this, we maintained DAF-2::degron animals on control plates and shifted them to 1 mM auxin-containing plates at day 0 (L4), and up to day 20 of adulthood (*Figure 5A*). We found that shifting the animals past the reproductive period at day 10 and day 12 still led to an increase in lifespan by 48–72% and 49–57%, respectively (*Figure 5A*, *Supplementary file 1*). Since transferring old *C. elegans* to culturing plates without bacterial food can also increase lifespan past reproduction (*Smith et al., 2008*), we decided to top-coat lifespan plates with auxin late in life. We observed lifespan extension of animals by supplementing auxin very late during lifespan at day 21 or 25 of adulthood, at a time at which already approximately 50–75% of the population had died (*Figure 5B–D*, *Supplementary file 1*). Remarkably, AID of DAF-2 doubled the lifespan of these animals at this late stage (*Figure 5A–D*, *Supplementary file 1*). For example, when about three-quarters of the population had ceased by day 21 (*Figure 5B*) or day 25 (*Figure 5D*), and control-treated DAF-2::degron animals lived for just another 4 or 7 days, the auxin-treated DAF-2::degron animals lived for another 26 or 43 days,

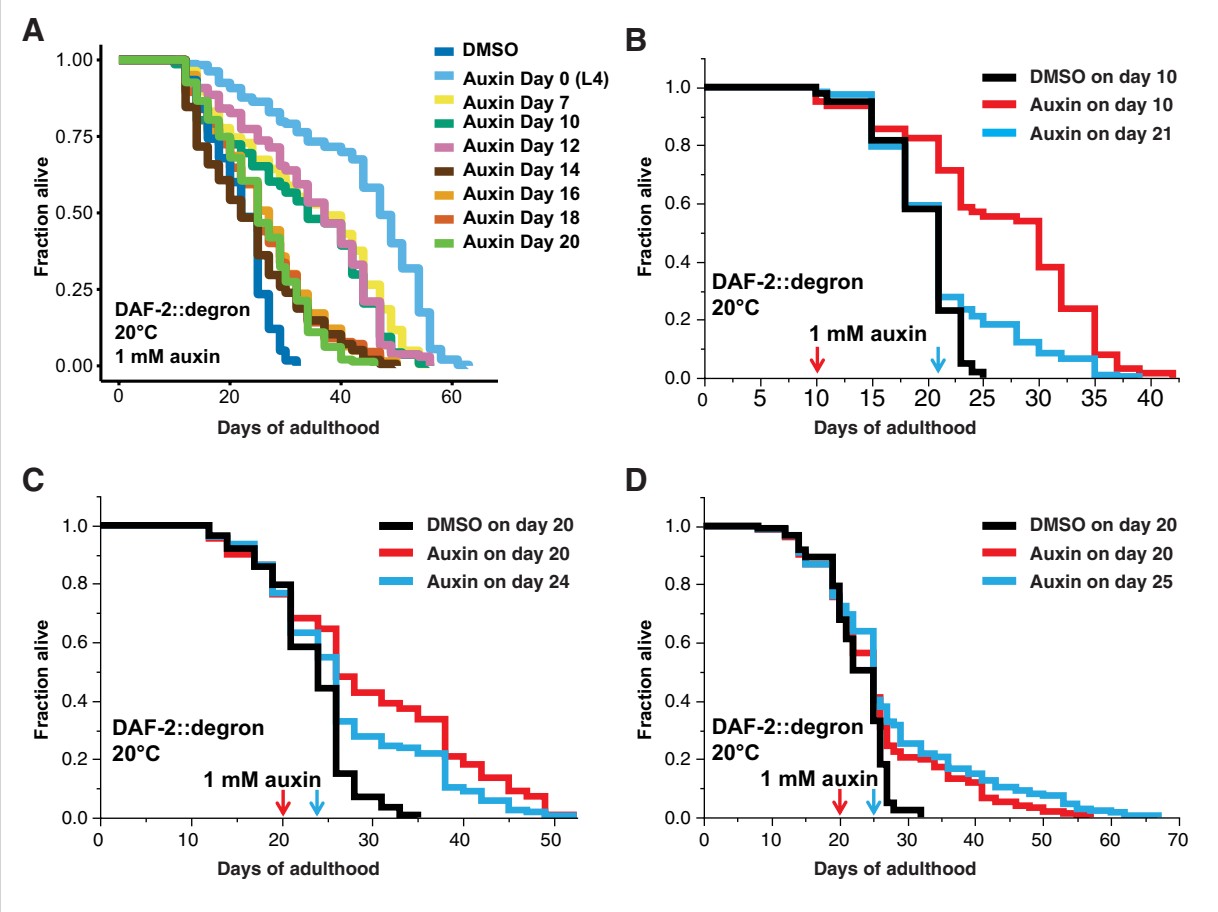

**Figure 5.** Late-life DAF-2 depletion extended lifespan. (**A**) Auxin treatment extended lifespan in aged DAF-2::degron animals. Animals were shifted from DMSO plates to plates containing 1 mM auxin on the indicated day. (**B–D**) Top-coating of plates with auxin to a final concentration of 1 mM, at the indicated days, extended the lifespan of DAF-2::degron animals during aging. For (A–D), see *Supplementary file 1* and *Source data 1* for raw data, statistics, and additional independent trials.

respectively (*Figure 5B and D*, *Supplementary file 1*). This demonstrates that reducing insulin/IGF-1 receptor signaling is feasible in geriatric *C. elegans*. Thus, our approach to selectively reduce DAF-2 protein in old animals suggests that targeting the *daf-2* signaling pathway late in life may be an effective strategy to extend lifespan.

## Discussion

For longevity interventions to be efficient without causing undesired side effects, the time point of treatment must be chosen carefully. This is especially important for pathways such as the insulin/IGF-1 pathway that is essential for growth and development (*Kenyon, 2010*). Although the importance of DAF-2 in regulating lifespan is well established, the consequences of late-in-life inhibition remained unknown.

Here, we demonstrate that late-life degradation of DAF-2 extends lifespan. In this work, we have effectively engineered a degron tag into the endogenous *daf-2* locus using CRISPR, representing the first report of auxin-induced degradation (AID) of a transmembrane receptor in vivo. DAF-2 receptor levels were strongly reduced via AID. Consistent with reduced insulin signaling, AID-mediated degradation of DAF-2 facilitated dauer formation, longevity, stress resistance, caused growth-related phenotypes during early adulthood, and finally increased longevity during the post-reproductive geriatric stages of life.

It is well established that DAF-2/insulin/IGF-1 receptor signaling connects nutrient levels to growth and development (*Murphy and Hu, 2013*). This is attributed to insulin-like peptides binding DAF-2

and activating a downstream phosphorylation kinase cascade that alters metabolism (*Murphy and Hu, 2013*). Surprisingly, we find that DAF-2 receptor abundance is linked to nutrient availability and perhaps dietary content. Starving *C. elegans* leads to decreased DAF-2 receptor abundance, consistent with previous in situ antibody staining (*Kimura et al., 2011*), whereas mimicking a high-energy diet by adding glucose to the bacterial food source increases DAF-2 receptor abundance. This suggests an additional layer of regulating IIS by connecting food cues to adapt metabolism via DAF-2 receptor levels, potentially as an internal representation of the environment.

Environmental conditions are carefully monitored by developing *C. elegans*. First, larval stage (L1) wild-type *C. elegans* that are food-deprived, exposed to elevated temperatures (>27 °C), and/or are crowded, enter into a pre-dauer L2d stage, where they continue monitoring their environment before committing and molting into dauers (*Hu, 2007*; *Karp, 2018*). However, previous temperature-shifting experiments (from 15°C to 25°C) with *daf-2* mutants have indicated the existence of a 'dauer decision time window' between the L1 and L2 stage (*Swanson and Riddle, 1981*). By using AID to manipulate DAF-2 levels directly, we found that AID degradation of DAF-2 during a narrow time period in the mid-L1 stage is sufficient to induce dauer formation, suggesting that the decision to enter dauer relies upon a threshold of DAF-2 protein levels. This decision is uncoupled from temperature or food abundance. Thus, absolute DAF-2 protein abundance appears to be a key factor in the animal's decision to enter into the dauer state during early development.

Although IIS is reduced in *daf-2(e1370* or *e1368)* mutants at lower temperatures (*Ewald et al., 2018*; *Gems et al., 1998*), dauer larvae are formed only when these *daf-2* mutants are exposed to higher temperatures. The observation that mutant DAF-2 protein (*Kimura et al., 2011*) but not wild-type DAF-2 protein (*Figure 5*) is lost at elevated temperatures suggests a model where DAF-2 mutant protein becomes unstable with increasing temperatures and might be subsequently targeted for degradation. This hypothesis might explain why strong class II *daf-2* mutants, such as *daf-2(e979)* and *daf-2(e1391)*, which have much lower DAF-2 protein levels at 15°C and 25°C compared to other *daf-2* mutants (*Tawo et al., 2017*), exhibit higher propensities toward dauer formation at any temperature (*Gems et al., 1998*). Thus, the severity of classical *daf-2* mutant alleles in regard to dauer formation and adult dauer traits might be linked to DAF-2 receptor abundance.

At higher temperatures, adult *daf-2* class II mutants exhibit significant penetrance of undesired phenotypes (*Ewald et al., 2018*). In these animals, a clear remodeling of the body and internal organs, including constriction of the pharynx and shrinkage of the germline, remains present, along with an altered neuronal morphology and electrical synapse connectome that drives behavioral changes such as quiescence, diminished foraging behavior, and altered egg-laying programs (*Arantes-Oliveira et al., 2003*; *Ewald et al., 2015*; *Gems et al., 1998*; *Hess et al., 2019*; *Bhattacharya et al., 2019*; *Patel et al., 2008*; *Podshivalova and Kerr, 2017*). We showed that L4-specific AID of the DAF-2::degron results in a non-conditional reduction of body size and brood size, whereas egg retention and germline shrinkage only occur at higher temperatures. This indicates that the non-conditional *daf-2* class II traits are not a residual effect of a dauer-like developmental program but instead are side effects caused by reduced functions of DAF-2 during later stages of development.

Additional temperature-sensitive *daf-2* class II traits of egg retention and gonad shrinkage are mediated by loss of *daf-2* in neurons only at higher temperatures. Why these traits only manifest at higher temperatures remains unclear. One explanation may be that DAF-2 levels are reduced in temperature-sensing neurons, which then elicits a systemic effect that drives germline shrinkage and egg retention. Neurons are refractory to RNAi, suggesting that *daf-2(RNAi)* effects work through other tissues than neurons to extend lifespan. Furthermore, treating class I mutants *daf-2(e1368)* with *daf-2(RNAi)* doubles their longevity without causing adult *daf-2* class II traits (*Arantes-Oliveira et al., 2003*), suggesting that *daf-2(RNAi)* would lower DAF-2 receptor levels in other tissues than neurons for this additive longevity effect.

Consistent with neuronal regulation of these traits is that *daf-2* RNAi applied to wild type does not result in dauers but results in dauer formation when applied to neuronal-hypersensitive RNAi *C. elegans* strains (*Dillin et al., 2002*; *Ewald et al., 2015*; *Kennedy et al., 2004*). We find that DAF-2 degradation in neurons or intestine increases lifespan. This is consistent with a previous finding by Apfeld and Kenyon that either mosaic loss of *daf-2* in AB cell lineage that gives rise to neurons and other cells (epidermis, seam, pharyngeal, vulval cells) or loss of *daf-2* in EMS cell lineage that gives rise to intestinal and other cells (pharyngeal and gonadal cells) results in increased lifespan (*Apfeld and*

*Kenyon, 1998*). By contrast, our neuronal DAF-2 depletion did not result in a doubling of lifespan as seen by loss of *daf-2* in the AB lineage (*Apfeld and Kenyon, 1998*), suggesting that loss of *daf-2* in other tissues, in combination with neurons or intestine, is required to recapitulate full lifespan extension. Given that reducing DAF-2 in neurons results in *daf-2* class II traits at higher temperatures, one might target intestinal DAF-2 for degradation to uncouple longevity from any undesired phenotypes. Yet, DAF-2 is essential for growth. We find the best time point for DAF-2 inhibition is rather late in life to bypass these undesired side effects to promote longevity.

We find that as late as day 25 of adulthood, when almost three-quarters of the population had died, AID of DAF-2 is sufficient to increase lifespan. The only other manipulation that was able to increase the lifespan so late in life was the transfer of old *C. elegans* to culture plates without food (*Smith et al., 2008*). However, *C. elegans* do not feed after reaching mid-life (*Collins et al., 2008*; *Ewald et al., 2016*), suggesting that it might not be the intake of calories that promotes longevity. Instead, the old *C. elegans* could sense the absence of food and thereby reduce DAF-2 levels to promote longevity. Along these lines, it would be interesting to determine if late-life bacterial deprivation works synergistically with DAF-2 AID or not in future studies.

In mammals, mid-life administration of IGF-1 receptor monoclonal antibodies to 78 -week-old mice (a time when all mice are still alive and 6 weeks before first mice start to die) is sufficient to increase their lifespan and improve their healthspan (*Mao et al., 2018*). Other parallels between mammals and nematodes are the tissues from which lower insulin/IGF-1 receptor levels promote longevity. Mice carrying brain-specific heterozygous IGF-1 receptor knockout are long-lived (*Kappeler et al., 2008*), as are the mice with adipose-specific knockout of the insulin receptor (*Blüher et al., 2003*). This is reminiscent of our findings that AID of DAF-2::degron in neurons or intestine (the major fat-storage tissue in *C. elegans*) was sufficient to increase the lifespan. These similarities could reflect conserved functions, as *daf-2* is considered the common ancestor to both insulin and IGF-1 receptors (*Kimura et al., 1997*).

Although it is known that neurons and the intestine are important for food perception and regulation of food intake, the effects of food perception or intake on insulin receptor and IGF-1 receptor levels are poorly understood. Starving rats for 3 days increases the abundance of insulin-bound insulin receptors (*Koopmans et al., 1995*), possibly to increase glucose uptake. It is unknown whether prolonged starvation would lead to lower basal insulin/IGF-1 receptor levels. However, alterations of the IGF-1 receptor levels are associated with altered lifespan. For instance, heterozygous IGF-1 receptor knockout mice, which have lower IGF-1 receptor levels, have an increased lifespan (*Holzenberger et al., 2003*; *Kappeler et al., 2008*; *Xu et al., 2014*). Also, overexpression of the short isoform of p53 (p44) increases IGF-1 receptor levels and shortens the lifespan of mice (*Maier et al., 2004*). Furthermore, the administration of recombinant human IGF-1 increases IGF-1 receptor abundance in murine embryonic cells (*Maier et al., 2004*). Food components themselves can affect insulin receptor levels. For instance, palmitate activates PPARα to induce *miR-15b,* which targets insulin receptor mRNA for degradation (*Li et al., 2019*). Potentially, food components could impact the murine insulin/IGF-1 receptor also via its degradation.

Indeed, there is some evidence suggesting the regulation of insulin/IGF-1 receptor abundance by E3 ubiquitin ligases. For instance, the E3 ligase CHIP regulates insulin/IGF-1 receptor levels in *C. elegans*, *Drosophila*, and human cell cultures (*Tawo et al., 2017*). In mice, the muscle-specific mitsugumin 53 (MG53) E3 ligase targets the insulin receptor for degradation (*Song et al., 2013*). High-fat diet results in the reduction of insulin receptor levels (*Li et al., 2019*) via higher MG53-mediated degradation (*Song et al., 2013*). MG53 is upregulated under a high-fat diet in mice, and MG53-/- deficient mice are protected from high-fat diet-induced obesity, insulin resistance, and other metabolic syndrome-associated phenotypes (*Song et al., 2013*). Furthermore, another E3 ligase, MARCH1, is overexpressed in obese humans and targets the insulin receptor for ubiquitin-mediated degradation (*Nagarajan et al., 2016*). Taken together, these observations suggest that food abundance controls mammalian insulin receptor levels via E3 ligase-mediated degradation. Although in *C. elegans* we found the opposite changes in DAF-2 receptor levels, that is, they were reduced upon starvation and increased upon high glucose feeding, our observations suggest that nutritional cues may regulate insulin/IGF-1 receptor levels via a variety of mechanisms, including ubiquitination and proteasomal degradation, across species.

In summary, we have demonstrated that interventions at almost the end of life can increase lifespan. We have established that auxin-induced degradation is suitable for targeting transmembrane receptors for non-invasive manipulations during developmental and longevity in vivo studies. We reconciled a longstanding question by providing evidence that dauer-like traits are not a spill-over of reprogrammed physiology from developing L2d pre-dauers. Instead, the essential growth-related functions of DAF-2 are causing deficits when applied during development or growth phases. We have shown that tissue-specific interventions or global interventions beyond reproduction or growth extend lifespan without pathology or deficits. Degradation of DAF-2/insulin/IGF-1 receptor might not be an artificial intervention since DAF-2/insulin/IGF-1 receptor abundance is read-out to adapt metabolism to food abundance. Dissecting intrinsic DAF-2/insulin/IGF-1 receptor abundance in response to nutritional cues may impact our understanding of nutrient sensing in promoting health during old age.

## Materials and methods

### Strains

All strains were maintained on NGM plates and OP50 *Escherichia coli* at 15 °C as described. The strains and primers used in this study can be found in *Supplementary file 3*.

### Statistical analysis and plotting

Statistical analysis was either done by using RStudio or Excel. All plots have been made using RStudio (1.2.5001). The packages ggplot2, survminer, dplyr were required for some plots.

### Auxin plates

Auxin (indole-3-acetic acid, Sigma #I3750) was dissolved in DMSO to prepare a 400 mM stock solution and stored at 4 °C. Auxin was added to NGM agar that has cooled down to about 60 °C before pouring the plates (*Zhang et al., 2015*). For lower concentrations (1 µM and 10 µM), the 400 mM stock dilution was further diluted in DMSO. Control plates contained the same amount of DMSO (0.25 % for 1 mM auxin plates). For lifespans, plates were supplemented with FUdR to the final concentration of 50 µM.

### Degron-tagged *Daf-2* strain and tissue-specific TIR1 expression

The sgRNA targeting the terminal exon of the annotated *daf-2* isoform a, 5' (G)TTTGGGGGTTTCAG-ACAAG 3' was cloned into the PU6:: sgRNA (*F* + E) plasmid backbone, pIK198 (*Katic et al., 2015*), yielding plasmid pIK323. The initial guanine was added to aid the transcription of the sgRNA. The underlined nucleotides in the sgRNA correspond to the stop codon of the DAF-2(A) protein.

The CRISPR tag repair template pIK325 degron::3xFLAG::SL2::SV40::NLS::degron::wrmscarlet::egl-13 NLS was assembled using the SapTrap method (*Schwartz and Jorgensen, 2016*) from the following plasmids: pMLS257 (repair template-only destination vector), pIK320 (wrmscarlet::syntron-embedded LoxP-flanked, reverse Cbr-unc-119), pMLS285 (*egl-13* NLS N-tagging connector), pIK321 (linker::auxin degron::3XFLAG::SL2 operon::SV40 NLS::linker::auxin degron C-tagging connector), and phosphorylated pairs of hybridized oligonucleotides oIK1182 5'TGGTCGGCTTTCGGTGAAAATGAGCATCTAATCGAGGATAATGAGCATCATCCACTTGTC 3', oIK1183 5'CGCGACAAGTGGATGATGCTCATTATCCTCGATTAGATGCTCATTTTCACCGAAAGCCGA 3', and oIK1184 5'ACGAACCCCCAAAAAATCCCGCCTCTTAAATTATAAATTATCTCCCACATTATCATATCT 3', oIK1185 5'TACAGATATGATAATGTGGGAGATAAATTTATAATTTAAGAGGCGGGATTTTTTGGGGGTT 3', respectively. Modules pIK320 and pIK321 (this study), compatible with the SapTrap kit, were assembled through a combination of synthetic DNA (Integrated DNA technologies) and molecular cloning methods. EG4322 *ttTi5605; unc-119(ed3)* animals were injected with a mix consisting of the sgRNA pIK323 at 65 ng/ml, tag repair template pIK325 at 50 ng/ml, pIK155 P*eft-3*::Cas9::*tbb-2* 3'UTR at 25 ng/ml and fluorescent markers pIK127 P*eft-3*::GFP::h2b::*tbb-2* 3'UTR at 20 ng/ml, and P*myo-3*::GFP at 10 ng/ml. Among the non-Unc F2 progeny of the injected animals not labeled with green fluorescence were correctly tagged *daf-2(bch40* [degron::3xFLAG::SL2::SV40 NLS::degron::wrmscarlet::*egl-13* NLS]) animals. We were able to recover two independent CRISPR alleles *daf-2(bch39)* and *daf-2(bch40)*.

pIK280 (TIR1::mRuby::*tbb-2* in a MosSCI-compatible backbone) (*Frøkjær-Jensen et al., 2012*; *Frøkjaer-Jensen et al., 2008*) was created by Gibson assembly (*Gibson et al., 2009*) from templates including pLZ31 (*Zhang et al., 2015*). Promoter regions were inserted by Gibson assembly of PCR

products into pIK280 to express TIR-1::mRuby in different tissues. Such plasmids were injected into EG4322 *ttTi5605; unc-119(ed3)* animals (*Frøkjær-Jensen et al., 2012*). The strains are IFM160 *bchSi59* [P*myo-3*::TIR1::mRuby::*tbb-2*] II; *unc-119(ed3)*, IFM161 *bchSi60* [P*vha-6*::TIR1::mRuby::*tbb-2*] II; *unc-119(ed3)*, and IFM164 *bchSi64* [P*rab-3*::TIR1::mRuby::*tbb-2*] II; *unc-119(ed3)*.

## Knockdown by RNA interference

RNAi bacteria cultures were grown overnight in LB with carbenicillin (100 µg/ml) and tetracycline (12.5 µg/ml), diluted to an $OD_{600}$ of 1, and induced with 1 mM IPTG and spread onto NGM plates containing tetracycline (12.5 µg/ml) and ampicillin (50 µg/ml) as described in *Ewald et al., 2017b*. Plasmid pL4440 was used as an empty RNAi vector (EV) control. The *daf-2(RNAi)* clone was a kind gift from the Blackwell lab and was sequenced for validation.

## Western blot

Synchronized *C. elegans*, on their first day of adulthood, were shifted to auxin plates for different time points. About 2000–5000 adult *C. elegans* per condition were disrupted using beads in lysis buffer (RIPA buffer [ThermoFisher #89900]), 20 mM sodium fluoride (Sigma #67414), 2 mM sodium ortho-vanadate (Sigma #450243), and protease inhibitor (Roche #04693116001) and kept on ice for 15 min before being centrifuged for 10 min at 15,000 × *g*. For equal loading, the protein concentration of the supernatant was determined with BioRad DC protein assay kit II (#5000116) and standard curve with Albumin (Pierce #23210). Samples were boiled at 37 °C for 30 min, shortly spun down, and 40 µg of protein was loaded onto NuPAGE Bis-Tris 10 % Protein Gels (ThermoFisher #NP0301BOX), and proteins were transferred to nitrocellulose membranes (Sigma #GE10600002). Western blot analysis was performed under standard conditions with antibodies against Tubulin (Sigma #T9026, 1:1000) (Sigma #F3165, 1:1000), FLAG-HRP (Sigma #A8592, 1:1000), and Degron (MBL #M214-3, 1:1000). HRP-conjugated goat anti-mouse (Cell Signaling #7076, 1:2000) secondary antibodies were used to detect the proteins by enhanced chemiluminescence (Bio-Rad #1705061). Quantification of protein levels was determined using ImageJ software and normalized to loading control (Tubulin). Statistical analysis was performed using either a two-tailed or one-tailed *t*-test. All western blots and quantifications can be found in *Source data 1* and *Source data 2*.

## Reporter assays

Transgenic *daf-16::gfp*; DAF-2::degron *C. elegans* were grown on plates for the indicated length of time supplemented with the corresponding concentration auxin at 20 °C. For image acquisition, the animals were placed on freshly made 2 % agar pads and anesthetized with tetramisole (*Teuscher and Ewald, 2018*). Images were taken with an upright bright-field fluorescence microscope (Tritech Research, model: BX-51-F) and a camera of the model DFK 23U × 236 (*Teuscher and Ewald, 2018*). For quantification, the animals were observed under a fluorescent stereomicroscope after the indicated amount of time has passed. *sod-3p::gfp* and *gst-4p::gfp* animals were incubated overnight at 20 °C and quantified the next morning. L4 larvae were used for quantification. Statistical analysis was performed by using Fisher's exact test for *daf-16::gfp* and *gst-4::gfp* and two-tailed *t*-test for *sod-3::gfp*. The DMSO control was compared to the ones treated with various concentrations of auxin.

## Developmental speed

As described in *Ewald et al., 2012*, L4 *C. elegans* of wild-type N2 and DAF-2::degron were picked to plates at 15 °C. After 2 days, the adult animals were shifted to new plates and were allowed to lay eggs for 2 hr. The stage of the offspring and their health was assayed 4 days later at 20 °C. Statistical analysis was performed by using a two-tailed *t*-test.

## SDS dauer assay

Synchronized L1 *C. elegans* were put on 1 mM auxin plates and incubated at 15 °C, 20 °C, and 25 °C. At the indicated time points, the animals were washed off with M9, shortly centrifuged down, and SDS was added for a final concentration of 1 %. After 10 min of gentle agitation, the animals were put on plates and checked for survival.

## Dauer pharynx

Adult *C. elegans* were placed on 1 mM auxin plates (for 'DAF-2::degron') or DMSO plates (for 'DAF-2::degron', *daf-2(e1368)* and *daf-2(e1370)*) and shifted to 25 °C. Dauer-like offspring or size-matching controls were picked after 4 days, anesthetized in 10 mM sodium azide, and mounted on 2 % agarose pads. Images were taken at 40× magnification using an inverted microscope (Tritech Research, MINJ-1000-CUST) and a camera of the model DFK 23U × 236.

## Feeding of fluorescent beads

Adult *C. elegans* were put on 1 mM auxin plates, or DMSO plate seeded with OP50 containing a 1:100 dilution of red fluorescent latex bead solution (Sigma #L3280) and shifted to 25 °C. Dauer offspring and control L2/L3 larvae on the bacterial lawn were picked after 4 days, anesthetized in 10 mM sodium azide, and mounted on 2 % agarose pads. An upright bright-field fluorescence microscope (Tritech Research, model: BX-51-F) and a camera of the model DFK 23U × 236 were used for image acquisition. The presence of beads in the intestine was checked at 20 × magnification. A two-tailed *t*-test was used for analysis.

## Dauer transition assay

Bleached eggs were synchronized for 2 days at 20 °C in an M9 buffer supplemented with 5 µg/ml cholesterol to yield a highly synchronous L1 population (*Teuscher et al., 2019*). The synchronized L1 larvae were put on OP50 NGM plates and then switched to plates containing 1 mM auxin at different time points. Dauer and non-dauer animals were counted after 2 days at 25 °C or 3 days at 20 °C.

## Gonadal cell count in L1

Bleached eggs were synchronized for 2 days at 20 °C in M9 buffer supplemented with 5 µg/ml cholesterol to yield a highly synchronous L1 population (*Teuscher et al., 2019*). The larvae were put on OP50 NGM plates for 24–25 hr, washed off and anesthetized with 0.25 mM tetramisole, and mounted on 2 % agarose pads. An upright bright-field fluorescence microscope (Tritech Research, model: BX-51-F) was used to count the cells in the developing gonad. Freshly hatched L1 have two germ stem cells (Z2-3), and L2 have, on average, 16 germ stem cells (*Mainpal et al., 2015*). However, we counted all visible cells in the gonad (i.e., both somatic and germ cells), which are four for the freshly hatched L1 (Z1-4) and 22 for animals around the L1/L2 transition (*Hubbard and Greenstein, 2000*). We relied on the characteristic round shapes of germ cells (*Altun and Hall, 2002*) (https://www.wormatlas.org/hermaphrodite/somatic%20gonad/Images/somaticfig3leg.htm). Using these criteria, we might have included DTC and other cells inside the gonad.

## Germline morphology and egg retention

L4 *C. elegans* maintained at 15 °C were picked on 1 mM auxin or control plates and shifted to the indicated temperatures. On the second day of adulthood, animals were mounted on 2 % agar pads and anesthetized with 0.25 mM tetramisole. Images were taken at 40× magnification on an inverted microscope (Tritech Research, MINJ-1000-CUST) and a camera of the model DFK 23U × 236. Statistical analysis was performed by using a two-tailed *t*-test for egg retention and Fisher's exact test for germline morphology.

## Body length measurement

L4 *C. elegans* maintained at 15 °C were picked on 1 mM auxin or control plates and shifted to the indicated temperatures. On the second day of adulthood, animals were mounted on 2 % agar pads and anesthetized with 0.25 mM tetramisole. Images were taken at 10 × magnification with an upright bright-field fluorescence microscope (Tritech Research, model: BX-51-F) and a camera of the model DFK 23U × 236. Body lengths were measured by placing a line through the middle of the body, starting from head to tail, using ImageJ 1.51 j. Statistical analysis was performed by using a two-tailed *t*-test.

## Progeny count

L4 *C. elegans* maintained at 15 °C were picked on 1 mM auxin or control plates and shifted to the indicated temperatures. *C. elegans* were shifted when necessary to fresh plates, and the progeny was

counted after 2 days of development. Animals that crawled off the plate, dug into the agar, or bagged precociously were censored. Statistical analysis was performed by using a two-tailed *t*-test.

## Lifespan assays

Synchronized L1 *C. elegans* were cultured at 15 °C or 20 °C on OP50 and shifted at the L4 stage to NGM plates containing 50 µM FUdR and auxin or DMSO. Bursted, dried out, or escaped animals were censored, and animals were considered dead when they failed to respond to touch and did not show any pharyngeal pumping. For late-life auxin lifespan assays: L4 *C. elegans* were picked onto NGM plates containing 50 µM FUdR. At day 20 or 25 of adulthood, plates were top-coated either DMSO or auxin to reach a final concentration of 0.25 % DMSO or 1 mM auxin with 0.25 % DMSO. Log-rank was used for statistical analysis. The plots were made by using the R-package survminer or JMP 14.1. All statistics can be found in *Supplementary file 1*.

## Arsenite assays

Oxidative stress assay was modified from *Ewald et al., 2017a*. *C. elegans* of the L1 or L4 stage were shifted to auxin or DMSO plates, washed off at the indicated time point, incubated with 5 mM sodium arsenite in U-shaped 96-well plates, and put into the wMicroTracker (MTK100) for movement scoring. For statistical analysis, the area under the curve was measured, and the mean for each run was calculated. Statistical analysis was performed by using a paired sample *t*-test. All plots can be found in *Supplementary file 2*.

## Acknowledgements

We thank Cyril Statzer for help with the analysis of the lifespan and oxidative stress data, Tea Kohlbrenner for help taking the alae pictures, Joy Alcedo and Benjamin Towbin for critical reading and feedback on the manuscript. Some strains were provided by the CGC, which is funded by NIH Office of Research Infrastructure Programs (P40 OD010440). RC was funded by the Research Council of Norway grant FRIMEDBIO-286499 and the EMBO Installation Grant No. 3615. The project POIR.04.04.00-00-203A/16 was carried out within the Team program of the Foundation for Polish Science, co-financed by the European Union under the European Regional Development Fund. CYE was funded by the Swiss National Science Foundation grant PP00P3_163898, and RV by the ETH Research Foundation Grant ETH-30 16–2.

## Additional information

### Funding

| Funder | Grant reference number | Author |
|---|---|---|
| Schweizerischer Nationalfonds zur Förderung der Wissenschaftlichen Forschung | PP00P3_163898 | Collin Yvès Ewald |
| Research Council of Norway | FRIMEDBIO-286499 | Rafal Ciosk |

The funders had no role in study design, data collection and interpretation, or the decision to submit the work for publication.

### Author contributions

Richard Venz, Conceptualization, Data curation, Formal analysis, Investigation, Methodology, Project administration, Performed all assays not listed by other authors. Wrote the manuscript with CYE in consultation with the other authors, Validation, Visualization, Writing – original draft; Tina Pekec, Investigation, Methodology, Crossed the DAF-2::degron strain and the transgenes into DAF-2::degron strain, Validation; Iskra Katic, Methodology, Writing – review and editing; Rafal Ciosk, Conceptualization, Funding acquisition, Writing – review and editing; Collin Yvès Ewald, Performed late-in-life

top-coating auxin lifespans. Wrote the manuscript with RV in consultation with the other authors, Conceptualization, Data curation, Formal analysis, Funding acquisition, Investigation, Methodology, Project administration, Resources, Supervision, Validation, Visualization, Writing – original draft, Writing – review and editing

### Author ORCIDs
Rafal Ciosk http://orcid.org/0000-0003-2234-6216
Collin Yvès Ewald http://orcid.org/0000-0003-1166-4171

### Decision letter and Author response
Decision letter https://doi.org/10.7554/eLife.71335.sa1
Author response https://doi.org/10.7554/eLife.71335.sa2

## Additional files

### Supplementary files
• Supplementary file 1. Lifespans of degron-tagged DAF-2. Trials that were performed in parallel are grouped together. (N) = number of animals observed; lifespan was measured from the L4 stage (see Materials and methods for details). Animals that left the plates, buried into the agar, bagged, or exploded were censored. L4440 empty vector was otherwise used as the control. The trial number is usually the starting date of the lifespan. All treatments were performed either starting from L4 or during adulthood, with DMSO and pL4440 empty vector plates used for rapamycin and RNAi controls, respectively. p-Values were obtained by the log-rank.

• Supplementary file 2. Auxin-inducible degradation (AID) DAF-2 phenotypes in comparison to *daf-2* mutant phenotypes. Our observed phenotypes loss or reduction of function with AID DAF-2::degron compared to literature reported phenotypes using *daf-2* mutants or *daf-2(RNAi)*. Studies are cited by using PMIDs.

• Supplementary file 3. Detailed strain list, primer, RNAi clone, and genomic sequences. Detailed *Caenorhabditis elegans* strain list with detailed genotype information. Sequencing results of RNAi clones used in this study. Primer sequences and sequencing results verifying genotypes or proper CRISPR editing.

• Transparent reporting form

• Source data 1. All raw numbers, quantifications, and statistics. Each tab corresponds to a figure or multiple figure panels as indicated. Provided are the raw values for making the graphs, the quantifications, and statistics.

• Source data 2. Original uncropped western blots. (A-I) Original uncropped western blots. The numbers next to the blot correspond to the trial date. (A–I)* Fully labeled original uncropped western blots. The numbers next to the blot correspond to the trial date. Each relevant band is labeled with the *Caenorhabditis elegans* strains and treatment conditions. The light blue dotted line corresponds to the cropped area. Additional information about the experimental conditions or to which figure or quantification the blot corresponds is indicated below the blot.

• Source data 3. Oxidative stress assays. Additional repeats and different experimental settings (indicated on the left of each row) for the oxidative stress assays using 5 mM sodium arsenite. Raw data and statistics are in *Source data 1*.

### Data availability
Source Data 1 includes data for all figures, Source Data 2 shows all full western blots and Source Data 3 contains raw data for all arsenite stress assays.

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
