## [Decision Letter]

**Acceptance summary:**

This paper by R. Venz et al., describes careful work involving the use of auxin-mediated degradation of the DAF-2 insulin/IGF-1 receptor in the nematode *Caenorhabditis elegans*. Since its initial discovery as a gene that could dramatically alter lifespan in this organism, daf-2 has been very extensively studied. Nevertheless, the authors make excellent and thorough use of their novel reagents to successfully add important new findings to our understanding of this broadly conserved aging pathway, including that intervening in the levels of this protein very late in life can still have dramatic effects on lifespan.

**Decision letter after peer review:**

Thank you for submitting your article "End-of-life targeted auxin-mediated degradation of DAF-2 Insulin/IGF-1 receptor promotes longevity free from growth-related pathologies" for consideration by *eLife*. Your article has been reviewed by 3 peer reviewers, and the evaluation has been overseen by a Reviewing Editor and Matt Kaeberlein as the Senior Editor. The following individual involved in review of your submission has agreed to reveal their identity: Joy Alcedo (Reviewer #3).

Essential revisions:

The reviewers and I are in agreement that the subject of the manuscript is of high interest, the data are high quality, and the assertions are generally well-supported by the data. There were a number of issues brought up in review, but a vast majority of them involve clarity and presentation of the data and we agree that they can all be addressed without additional experiments. I have consolidated the reviewer comments below, as addressing them all should prove feasible and will improve the text.

1) An important claim made in the abstract and several times in the paper is that depleting DAF-2 late in life increases lifespan while avoiding "dauer-associated" phenotypes associated with earlier depletion of DAF-2. The increase in lifespan is shown in figure 5, but "dauer-associated" phenotypes are not addressed in worms with this treatment. Please clarify what you mean by dauer-associated phenotypes. To many, dauer-associated phenotypes involve phenotypes acquired during the dauer state. If the animals never entered the dauer state, then any subsequent phenotypes are not the result of the dauer state.

2) Another claim that is emphasized is regarding the time when DAF-2 activity is required to promote non-dauer development. There are three aspects related to this claim that require more explanation: 1) statements are made on Line 264 and line 464 that reducing daf-2 during the mid-L1 stage causes dauer formation irrespective of conditions later in development. However, data were not presented addressing the environment at L2d or later stages, other than the observation made earlier that dauer formation is induced at all temperatures tested, 2) The statement on line 465 that the abundance of DAF-2 appears to be the key or sole factor in the dauer formation decision should be revised to reflect the fact that there are several interconnected dauer formation pathways and mutations in genes in any of these pathways are sufficient to cause dauer formation. There is no indication that daf-2 is more important than daf-7 or daf-12, for example, 3) experiments shown in supplemental figure 3 address the precise timing when daf-2 activity is needed to promote non-dauer development. There are three different criteria used to describe the stages in question. The text describes animals as mid-L1. Supplemental figures 1A-B show hours since the onset of feeding. And finally supplemental figure 3C shows the number of germ cells per animal. However, these three criteria are never put in context relative to each other, so it is difficult to understand how they all relate to each other. Under the conditions in the experiments, when does the L1 molt occur in terms of hours post-feeding? How many germ cells are present at different stages of the L1? Is the correlation between germ cell number and developmental stage based on your own observations or previously published work?

3) There are several places where the information in figure legends and/or methods is incomplete. Some examples are given below, but in general the authors are encouraged to revisit the legends and methods section and make sure the experiments are clearly described.

Examples of this include:

A. In some figures when statistical significance is shown as asterisks over a single bar in a graph, it is not clear which samples are being compared. Adding bars to show the comparisons or information in the figure legend would clarify this point.

B. Please add methods to explain how the starvation experiments shown in figures 1F-1H were performed.

C. Please add methods to explain how RNAi experiments were performed.

D. What stage animals were used for the reporter experiments used in figure 2?

E. In figure 2F, at what stage are the arrested animals? From what I can see they don't look like dauers.

F. In supplementary figure 3, how were germ cells recognized and distinguished from cells in the somatic gonad?

G. In supplementary figure 4A-B, which specific genotypes are shown?

H. In figure 5, panel D is not mentioned in the legend. A description of panel D should be added, together with an explanation of how panel C and panel D differ from each other, since they seem to be showing very similar information.

4) Similarly, communication of information would be improved by the following:

A. The text within figures is incredibly tiny. All of the figures would benefit from dramatically increasing the size of the text to fill more of the white space available.

B. In supplemental figure 1D, without reading the methods section, it is not clear what the experiment is that is being done here. Consider adding more information to the figure legend.

C. It would be helpful to have brief explanation of why phenotypes such as egg-retention are called "dauer-associated", when dauer larvae are too young to have eggs. Reduced motility and stress resistance are examples of phenotypes that are shared by dauers and daf-2 mutant adults, but the reproductive phenotypes seem different.

5) A more minor, but very interesting claim made on line 180 concerns the effect of the environment on DAF-2 levels. Building on the results of Kimura et al., the authors show two different experiments, one in which starvation causes a dramatic reduction in DAF-2 levels (Figure 1F-1G) and a second experiment in which no effect on DAF-2 levels was seen (Figure 1H). They interpret the difference between these experiments to be the different bacterial strains that worms were fed prior to starvation. However, there appears to be many differences between these two experiments. The OP50 experiments were presumably carried out on standard NGM plates, since no additional information was provided. In contrast, the HT1115 experiments mention L4440, which is the empty vector control used for RNAi. This presumably means that worms were exposed to RNAi conditions, including IPTG and ampicillin, as well as the double-stranded RNAi coming from the empty vector. Furthermore, FUDR was mentioned for this experiment but not the OP50 experiment. Therefore, in principle any one of these conditions, or a combination, could have affected DAF-2 levels. If the authors wish to make conclusions about the bacterial strain they should perform an experiment where the bacterial strain is the only difference in experimental conditions. Alternatively, they can adjust their wording to encompass all of the differences between the experiments. In any case, the experimental details for these experiments should be added to the methods section.

6) In Supplementary Figure 3A and 3B, please annotate the timepoints associated with the L1 and L2 molts in their graphs.

---

## [Author Response]

Essential revisions:The reviewers and I are in agreement that the subject of the manuscript is of high interest, the data are high quality, and the assertions are generally well-supported by the data. There were a number of issues brought up in review, but a vast majority of them involve clarity and presentation of the data and we agree that they can all be addressed without additional experiments. I have consolidated the reviewer comments below, as addressing them all should prove feasible and will improve the text.

We thank the reviewers and the editor for the constructive and clear feedback.

1) An important claim made in the abstract and several times in the paper is that depleting DAF-2 late in life increases lifespan while avoiding "dauer-associated" phenotypes associated with earlier depletion of DAF-2. The increase in lifespan is shown in figure 5, but "dauer-associated" phenotypes are not addressed in worms with this treatment. Please clarify what you mean by dauer-associated phenotypes. To many, dauer-associated phenotypes involve phenotypes acquired during the dauer state. If the animals never entered the dauer state, then any subsequent phenotypes are not the result of the dauer state.

We thank the reviewers for pointing out this flaw in logic. The term dauer-like has been used by our and other labs (PMID: 14576426, 15308663, 28934747) before describing these daf-2 mutant phenotypes. We realized that “dauer-associated” is too vague. What we wanted to define with this term were the daf-2 class II mutant phenotypes that emerge at higher temperatures during adulthood, as characterized by David Gems (PMID: 9725835). Thus, we changed most of the “dauer-associated” to “daf-2 class II mutant” phenotypes. To maintain the flow of the text, we replaced the remaining few “dauer-associated” terms with “dauer-like”.

We have also rewritten the abstract, since as pointed out in Figure 5, we do not provide quantitative data measuring these “dauer-associated” phenotypes. We only observed that these geriatric AID daf-2 animals are mobile (i.e., do not show dauer-like quiescence).

2) Another claim that is emphasized is regarding the time when DAF-2 activity is required to promote non-dauer development. There are three aspects related to this claim that require more explanation: 1) statements are made on Line 264 and line 464 that reducing daf-2 during the mid-L1 stage causes dauer formation irrespective of conditions later in development. However, data were not presented addressing the environment at L2d or later stages, other than the observation made earlier that dauer formation is induced at all temperatures tested,

We agree, the statements are based on indirect observation and would need more direct evidence. Thus, we deleted this claim and changed the text to:

For Line 264: “We found that when DAF-2 levels are below a given threshold at the mid-L1 stage, the animals commit to dauer formation.”

For Line 464: “This decision is uncoupled from temperature or food abundance.”

2) The statement on line 465 that the abundance of DAF-2 appears to be the key or sole factor in the dauer formation decision should be revised to reflect the fact that there are several interconnected dauer formation pathways and mutations in genes in any of these pathways are sufficient to cause dauer formation. There is no indication that daf-2 is more important than daf-7 or daf-12, for example,

The reviewers are correct. We rewrote this statement to:

“Thus, absolute DAF-2 protein abundance appears to be a key factor in the animal’s decision to enter into the dauer state during early development.”

3) Experiments shown in supplemental figure 3 address the precise timing when daf-2 activity is needed to promote non-dauer development. There are three different criteria used to describe the stages in question. The text describes animals as mid-L1. Supplemental figures 1A-B show hours since the onset of feeding. And finally supplemental figure 3C shows the number of germ cells per animal. However, these three criteria are never put in context relative to each other, so it is difficult to understand how they all relate to each other. Under the conditions in the experiments, when does the L1 molt occur in terms of hours post-feeding? How many germ cells are present at different stages of the L1? Is the correlation between germ cell number and developmental stage based on your own observations or previously published work?

In supplementary Figure 1A-B, DAF-2degron animals were bleached and eggs were synchronized for two days at 20°C in an M9 buffer supplemented with 5 μg/ml cholesterol to yield a highly synchronous L1 population. The synchronized L1 larvae were put on OP50 NGM plates and then switched to plates containing 1 mM auxin at different time points. Dauer and non-dauer animals were counted after two days at 25°C or three days at 20°C.

Given that this experimental procedure does not give any information on the larval stage at which DAF-2 AID is effective in inducing dauers, we used the same procedure and determined the larval stage by counting the germ cells. Based on the timing of about 24-25 hours, 50% of the population became dauer vs non-dauer; we determined the larval stage at this time point by counting the germ cells. Freshly hatched L1 have 2 germ stem cells (Z2-3) and L2 have on average 16 germ stem cells (PMID: 26395476). However, we counted all visible cells in the gonad, which are 4 for freshly hatched L1 (Z1-4) and 22 for L1/L2 transition (PMID: 10822256). We relied on the characteristic round shapes of germ cells

(https://www.wormatlas.org/hermaphrodite/somatic%20gonad/Images/somaticfig3leg.htm). However, we might have included DTC and other cells inside the gonad. Thus, we revised the text to “gonadal cells” and updated the Materials and methods.

3) There are several places where the information in figure legends and/or methods is incomplete. Some examples are given below, but in general the authors are encouraged to revisit the legends and methods section and make sure the experiments are clearly described.

We revisited, updated, and revised figure legends and methods section.

Examples of this include:A. In some figures when statistical significance is shown as asterisks over a single bar in a graph, it is not clear which samples are being compared. Adding bars to show the comparisons or information in the figure legend would clarify this point.

We revisited the figures and stated or indicated more clearly which groups were statistically compared.

B. Please add methods to explain how the starvation experiments shown in figures 1F-1H were performed.

We updated the figure legend with more details.

A representative immunoblot analysis of “DAF-2::degron” animals after 1% glucose and 36 hours to 48 hours starvation on the second day of adulthood. L4 DAF-2::degron animals were either placed on OP50 NGM plates with or without 1 mM auxin, or containing 1% glucose, or on empty (no bacteria) NGM plates. Animals were harvested 36-48 hours later.

C. Please add methods to explain how RNAi experiments were performed.

We added a paragraph “Knockdown by RNA interference” in the Method section.

D. What stage animals were used for the reporter experiments used in figure 2?

Fig-2B were L4 and Figure 2C-D were day 1 adults (L4 + 24 hours). We updated the figure legends for clarity.

E. In figure 2F, at what stage are the arrested animals? From what I can see they don't look like dauers.

In Figure 2F at 100 μm auxin these animals are pre-dauer and as indicated in the figure legend for 2G will develop into dauers a couple of days later.

F. In supplementary figure 3, how were germ cells recognized and distinguished from cells in the somatic gonad?

As indicated above, we relied on the characteristic round shapes of germ cells (https://www.wormatlas.org/hermaphrodite/somatic%20gonad/Images/somaticfig3leg.htm), however, we might have included DTC and other cells inside the gonad. We have revised the text accordingly and updated the Materials and methods.

G. In supplementary figure 4A-B, which specific genotypes are shown?

We updated the figure legend with the exact genotype and treatment.

Supplementary Figure 4

(A) Representative images of the egg retention phenotype. Left image is DAF-2::degron treated with DMSO. Right image is daf-2(e1370). The asterisk indicates eggs that were retained in the germline. Bar = 100 μm

(B) Representative image of the different levels of germline sizes. Top image DAF-2::degron with DMSO. Middle image DAF-2::degron treated with 1 mM auxin. Bottom image is daf-2(e1370). Bar = 100 μm

H. In figure 5, panel D is not mentioned in the legend. A description of panel D should be added, together with an explanation of how panel C and panel D differ from each other, since they seem to be showing very similar information.

It should have read (B-D) not (B-C). D is the same experimental setup as B and C to show 3 independent trials with the top-coating auxin or DMAs method.

4) Similarly, communication of information would be improved by the following:A. The text within figures is incredibly tiny. All of the figures would benefit from dramatically increasing the size of the text to fill more of the white space available.

We went through all figures again and increased the text and sometimes rearranged the panels to reduce the white space.

B. In supplemental figure 1D, without reading the methods section, it is not clear what the experiment is that is being done here. Consider adding more information to the figure legend.

The legend for Supplementary Figure 1D was updated with additional information.

“Comparison of the developmental speed of wild type (N2) and DAF-2::degron from n = 3 independent experiments. A timed egg lay for 2 hours for synchronization and after 4 days at 20°C, the developmental stage was scored. Error bar represents s.d.”

C. It would be helpful to have brief explanation of why phenotypes such as egg-retention are called "dauer-associated", when dauer larvae are too young to have eggs. Reduced motility and stress resistance are examples of phenotypes that are shared by dauers and daf-2 mutant adults, but the reproductive phenotypes seem different.

As mentioned above, we have replaced “dauer-associated” with “daf-2 class II mutant” for these phenotypes.

5) A more minor, but very interesting claim made on line 180 concerns the effect of the environment on DAF-2 levels. Building on the results of Kimura et al., the authors show two different experiments, one in which starvation causes a dramatic reduction in DAF-2 levels (Figure 1F-1G) and a second experiment in which no effect on DAF-2 levels was seen (Figure 1H). They interpret the difference between these experiments to be the different bacterial strains that worms were fed prior to starvation. However, there appears to be many differences between these two experiments. The OP50 experiments were presumably carried out on standard NGM plates, since no additional information was provided. In contrast, the HT1115 experiments mention L4440, which is the empty vector control used for RNAi. This presumably means that worms were exposed to RNAi conditions, including IPTG and ampicillin, as well as the double-stranded RNAi coming from the empty vector. Furthermore, FUDR was mentioned for this experiment but not the OP50 experiment. Therefore, in principle any one of these conditions, or a combination, could have affected DAF-2 levels. If the authors wish to make conclusions about the bacterial strain they should perform an experiment where the bacterial strain is the only difference in experimental conditions. Alternatively, they can adjust their wording to encompass all of the differences between the experiments. In any case, the experimental details for these experiments should be added to the methods section.

We agree that for this statement, more research is required to dissect this. Thus, we changed the text accordingly.

“Curiously, we noted that this starvation-induced degradation of DAF-2 did not happen when, during development, *C. elegans* were fed another bacterial strain, HT1115, used for RNAi (L4440). When DAF-2::degron animals were grown on L4440 and then shifted on empty NGM plates for 24 or 48 hours of starvation, DAF-2 levels did not decrease (Figure 1H). This observation suggests a hypothesis that the nutritional composition of the animal’s diet prior to starvation influences DAF-2 stability, which will be interesting to test in future research. We conclude that food availability controls not only the secretion of insulin-like peptides to regulate DAF-2 activity (Pierce et al., 2001), but also DAF-2 receptor abundance.”

We updated the method section and added experiment-specific details into the corresponding figure legends.

6) In Supplementary Figure 3A and 3B, please annotate the timepoints associated with the L1 and L2 molts in their graphs.

As indicated in Supplementary Figure 3C, after 24-25 hours of starved L1 refeeding, animals are still in L1. Thus, in Sup. Figure 3A-B, these are all presumably L1 staged animals.